# Survival Permanental Processes for Survival Analysis with Time-Varying Covariates

**Hideaki Kim**
NTT Human Informatics Laboratories
NTT Corporation
`hideaki.kin@ntt.com`

## Abstract

Survival or time-to-event data with time-varying covariates are common in practice, and exploring the non-stationarity in covariates is essential to accurately analyzing the nonlinear dependence of time-to-event outcomes on covariates. Traditional survival analysis methods such as Cox proportional hazards model have been extended to address the time-varying covariates through a counting process formulation, although sophisticated machine learning methods that can accommodate time-varying covariates have been limited. In this paper, we propose a non-parametric Bayesian survival model to analyze the nonlinear dependence of time-to-event outcomes on time-varying covariates. We focus on a computationally feasible Cox process called *permanental process*, which assumes the square root of hazard function to be generated from a Gaussian process, and tailor it for survival data with time-varying covariates. We verify that the proposed model holds with the representer theorem, a beneficial property for functional analysis, which offers us a fast Bayesian estimation algorithm that scales linearly with the number of observed events without relying on Markov Chain Monte Carlo computation. We evaluate our algorithm on synthetic and real-world data, and show that it achieves comparable predictive accuracy while being tens to hundreds of times faster than state-of-the-art methods.

## 1 Introduction

Survival or time-to-event data analysis has been widely applied for analyzing the dependence of survival time, the time until an event occurs, on covariates. They have a wide ranging list of applications in reliability engineering [27], finance [14], marketing [20], and especially, clinical research [4, 5, 28].

An essential part of the survival data analysis is to estimate a *hazard function*, that is, the instantaneous probability of events occurring at any particular time, as a regression function of covariates. Given a hazard function, we can evaluate the impact of covariates on survival times, and assess the degree of risk of experiencing an event in the future, through the survival function, i.e., the probability of no events occurring during a specified interval. The literature contains a vast number of studies that have proposed survival models to estimate hazard functions, most of which assume that covariates are time-invariant (e.g., gender and age at the time of diagnosis in clinical research): they range from the classical Cox proportional hazards model [5] to modern machine learning models based on generalized boosting machines [38], random forests [17], Gaussian processes [11, 25], and deep neural networks [10, 22, 49]. However, time-varying covariates are common in survival data, and the rapid advances in data collection allow us to access long-term and high temporal resolution covariate data. This is encouraging researchers to explore the non-stationarity in covariates for the accurate estimation/prediction of time-to-event outcomes. An important application of sur-

vival analysis with time-varying covariates is what-if analysis: through simulations of future events under various covariate functions over time, we can find the optimal covariate function or policy that would yield a desirable future state. For example, in reliability engineering applications where the events are the failures of machines, time-varying covariates could be the maintenance schedule and the room temperature/humidity controlled by air conditioners. The maintenance manager can optimize the schedules of maintenance and air conditioning by balancing the risk of failure and the costs of maintenance and air conditioning. If survival models for static covariates were applied to non-stationary data, then the static survival models would fail to estimate the underlying dependence of the hazard function on covariates, resulting in unreliable decision-making. Therefore, survival models that accommodate time-varying covariates are needed. This paper assesses survival data with time-varying covariates to estimate a hazard function that takes covariates into account.

The standard survival model is the Cox proportional hazards model [5, 6, 40] (CoxPH), which forms the logarithm of the hazard function as a linear combination of covariates. The original CoxPH assumes that covariates are time-invariant and have a constant log-linear effect over time on the hazard function. It was extended to address time-varying covariates through the technique of counting process formulation [1]. Although the extended CoxPH is very efficient to compute, it fails to address the nonlinear dependence of survival times on covariates, which is often the case in real-world data. To rectify this limitation, two non-parametric machine learning models have been proposed: generalized boosted models [16] and random forest-based models [47, 48], both of which adopt the extended CoxPH as a learner/tree, and construct non-parametric predictors in an ensemble manner. They succeeded in addressing the nonlinear dependence of survival times on time-varying covariates, but the large number of trees needed for accurate predictions tend to make the algorithm too slow and ineffective for real-time applications.

In this paper, we propose a novel Bayesian survival model to estimate a hazard function as a nonlinear regression function of covariates from survival data with time-varying covariates. To construct a scalable algorithm, we utilize *permanental process*, a doubly-stochastic point process which assumes the square root of hazard function to be generated from a Gaussian process; its computational advantages have been highlighted by recent studies [12, 18, 24, 30, 45]. We tailor it for survival data in the counting process format, and show that the tailored process, which we call the *survival permanental process*, holds the representer theorem [44]: the maximum a posteriori estimator of the latent function can be represented as a finite linear combination of a transformed kernel product evaluated on the event time points. The representer theorem offers us a feasible Bayesian estimation algorithm that scales linearly with the number of events observed without relying on Markov Chain Monte Carlo computation. Furthermore, we derive the predictive distribution and the marginal likelihood in one feasible form, enabling us to implement the uncertainty evaluation and the hyper-parameter optimization in a Bayesian manner. To the best of our knowledge, it is the first time to exploit the representer theorem for survival data analysis. We provide python codes to reproduce the results in this paper[1].

In Section 2, we introduce the survival permanental process (SurvPP) and construct a scalable Bayesian estimation algorithm for it. In Section 3, we outline related work. In Section 4, we compare SurvPP with reference survival models on synthetic and real-world data, and confirm that our algorithm achieves comparable predictive accuracy while being tens to hundreds of times faster than state-of-the-art survival models. Finally, Section 5 states our conclusions.

## 2 Methods

### 2.1 Survival Permanental Processes

We assume that a right-censored dataset with time-varying covariates, $\mathcal{D} = \{(\boldsymbol{y}_u(t), T_u, \eta_u)\}_{u=1}^U$, is observed, where $\boldsymbol{y}_u(t) : (0, T_u] \to \mathcal{Y} \subset \mathbb{R}^d$, $T_u \in \mathbb{R}$, and $\eta_u \in \{0, 1\}$ are the map of $d$-dimensional covariate, the end time of observation, and the indicator that represents whether an individual experienced an event ($\eta_u = 1$) or was right-censored ($\eta_u = 0$) at $T_u$, for individual $u \in \{1, 2, \dots, U\}$, respectively. We consider tailoring a permanental process for the survival data, where latent function, $x(\boldsymbol{y}) : \mathcal{Y} \to \mathbb{R}$, is generated from a Gaussian process (GP) on covariate space $\mathcal{Y}$, and an event time for each individual $u$ is generated from a point process with hazard function, $\lambda_u(t)$,

---

[1]Code and data are provided at `https://github.com/HidKim/SurvPP`.

such that

$$p(x(\boldsymbol{y})|\mathcal{D}) = \frac{p(\mathcal{D}|x(\boldsymbol{y}))\,\mathcal{GP}(x(\boldsymbol{y})|k)}{\int \mathscr{D}x(\boldsymbol{y})\,p(\mathcal{D}|x(\boldsymbol{y}))\,\mathcal{GP}(x(\boldsymbol{y})|k)}, \tag{1}$$

$$\log p(\mathcal{D}|x(\boldsymbol{y})) = \sum_{u=1}^{U}\left[\eta_u \log \lambda_u(T_u) - \int_0^{T_u}\lambda_u(t)dt\right], \quad \lambda_u(t) = x^2(\boldsymbol{y}_u(t)), \tag{2}$$

where $\mathcal{GP}(x(\boldsymbol{y})|k)$ represents a GP with kernel function $k(\boldsymbol{y},\boldsymbol{y}')$, $p(\mathcal{D}|x(\boldsymbol{y}))$ is the likelihood function of the point process, while $\int \mathscr{D}x(\boldsymbol{y})$ in the denominator represents the integral over the function or the infinite-dimensional variable $x(\boldsymbol{y})$. We call the model defined by (1-2) the *survival permanental process* (SurvPP). Hazard function $\lambda_u(t)$ represents an instantaneous probability of events occurring at each point in time, and our goal is to estimate the functional form of the hazard function over covariate domain, $x^2(\boldsymbol{y})$, from the survival data $\mathcal{D}$.

Given a hazard function over covariate domain, $x^2(\boldsymbol{y})$, and a covariate map for individual $u$, $\boldsymbol{y}_y(t)$, we can evaluate the survival function and the probability density distribution of event time at arbitrary time point $t$, denoted by $S(t)$ and $P_e(t)$, respectively, as follows:

$$S(t) = \exp\left[-\int_0^t x^2(\boldsymbol{y}_u(s))ds\right], \quad P_e(t) = x^2(\boldsymbol{y}_u(t))\exp\left[-\int_0^t x^2(\boldsymbol{y}_u(s))ds\right]. \tag{3}$$

By using (3), we can perform survival analyses that include the analysis of the expected duration of event time, and the assessment of the degree of risk that an individual will experience the event before a specified period.

## 2.2 Counting Process Format of Data

Traditionally, survival data with time-varying covariates have taken the *counting process format* [1]. The format assumes that, for each individual $u$, covariates are measured at $J_u$ finite representative time points and can be regarded as constant between successive time points,

$$\boldsymbol{y}_u(t) = \boldsymbol{y}_u^j, \quad t \in (s_u^j,\ s_u^{j+1}], \quad j = 0,\ldots,J_u - 1, \tag{4}$$

where $(s_u^0, s_u^{J_u}) = (0, T_u)$. It then splits individual $u$'s record, $(\boldsymbol{y}_u(t), T_u, \eta_u)$, into $J_u$ pseudo-individual records as in

$$\{(s_u^j,\ s_u^{j+1},\ \xi_u^j,\ \boldsymbol{y}_u^j)\}_{j=0}^{J_u-1}, \quad \xi_u^j = \eta_u \cdot \mathrm{I}(j = J_u - 1), \tag{5}$$

where $\mathrm{I}(\cdot)$ represents the indicator. Joining the pseudo-individual records of $U$ individuals together results in the counting process format of data:

$$\mathcal{D} = \{(T_j^0,\ T_j^1,\ \xi_j,\ \boldsymbol{y}_j)\}_{j=1}^{J}, \quad J = \sum_{u=1}^{U} J_u. \tag{6}$$

In this paper, we employ the counting process format of (6), and rewrite the likelihood function of SurvPP (2) as

$$\log p(\mathcal{D}|x(\boldsymbol{y})) = \sum_{j=1}^{J}\left[\xi_j \log x^2(\boldsymbol{y}_j) - (T_j^1 - T_j^0)x^2(\boldsymbol{y}_j)\right] = \sum_{n=1}^{N}\log x^2(\tilde{\boldsymbol{y}}_n) - \sum_{j=1}^{J}\Delta_j x^2(\boldsymbol{y}_j) \tag{7}$$

where $\Delta_j$ and $\{\tilde{\boldsymbol{y}}_n\}_{n=1}^{N}$ represent the durations of observation and the covariates observed at event times, respectively:

$$\Delta_j = T_j^1 - T_j^0, \quad \{\tilde{\boldsymbol{y}}_n\}_{n=1}^{N} = \{\boldsymbol{y}_j|\eta_j = 1, 1 \le j \le J\}, \quad N = \sum_{j=1}^{J}\xi_j = \sum_{u=1}^{U}\eta_u. \tag{8}$$

Note that the number of events, $N$, is usually much smaller than the number of pseudo-individual records: $N \ll J$.

In practice, there are two ways of defining the representative time points $\{s_u^j\}_{j=0}^{J_u}$: one is as the points that were observed, which might be sparse over time due to measurement constraints; the other is as the denser points obtained by using interpolation methods (e.g., splines). This paper assumes the latter, and thus various values of $J_u$ were considered in the synthetic data experiment (see Section 4). Although the counting process format of input assumes piecewise stationarity in covariates, this assumption is not so strong because we can adopt the representative time points in any density if necessary.

## 2.3 Maximum *A Posteriori* Estimator

We consider the problem of obtaining the maximum *a posteriori* (MAP) estimator of $x(\boldsymbol{y})$, denoted by $\hat{x}(\boldsymbol{y})$, to maximize the posterior probability (1). We derive $\hat{x}(\boldsymbol{y})$ through the approach of using the path integral representation of GP [23],

$$\mathcal{GP}(x(\boldsymbol{y})|k)\,\mathscr{D}x(\boldsymbol{y}) = \sqrt{\frac{1}{|\mathcal{K}|}}\exp\left[-\frac{1}{2}\iint_{\mathcal{Y}\times\mathcal{Y}}k^*(\boldsymbol{y},\boldsymbol{y}')x(\boldsymbol{y})x(\boldsymbol{y}')d\boldsymbol{y}d\boldsymbol{y}'\right]\mathscr{D}x(\boldsymbol{y}), \quad (9)$$

where $\mathcal{K}$ and $\int_{\mathcal{Y}}k^*(\boldsymbol{y},\boldsymbol{y}')\cdot d\boldsymbol{y}'$ are the integral operator with kernel function $k(\boldsymbol{y},\boldsymbol{y}')$ and its inverse operator, respectively: $\mathcal{K}^{(*)}x(\boldsymbol{y}) = \int_{\mathcal{Y}}k^{(*)}(\boldsymbol{y},\boldsymbol{y}')x(\boldsymbol{y}')d\boldsymbol{y}'$, $\mathcal{K}k^*(\boldsymbol{y},\boldsymbol{y}') = \delta(\boldsymbol{y}-\boldsymbol{y}')$. Here $|\mathcal{K}|$ represents the function determinant [15] of $\mathcal{K}$, defined by the product of its eigenvalues [23]. Using the representation of (9), we write the posterior of SurvPP (1-2) in the following functional form,

$$p(x(\boldsymbol{y})|\mathcal{D})\mathscr{D}x = \frac{1}{p(\mathcal{D})}\exp\left[-S\big(x(\boldsymbol{y}),\underline{x}(\boldsymbol{y})\big) - \frac{1}{2}\log|\mathcal{K}|\right]\mathscr{D}x, \quad (10)$$

where $S\big(x(\boldsymbol{y}),\underline{x}(\boldsymbol{y})\big)$ is the *action integral*, defined by

$$S\big(x(\boldsymbol{y}),\underline{x}(\boldsymbol{y})\big) = \int_{\mathcal{Y}}\left[\frac{1}{2}x(\boldsymbol{y})\underline{x}(\boldsymbol{y}) + \sum_{j=1}^{J}x^2(\boldsymbol{y})\Delta_j\delta(\boldsymbol{y}-\boldsymbol{y}_j) - 2\sum_{n=1}^{N}\log|x(\boldsymbol{y})|\delta(\boldsymbol{y}-\tilde{\boldsymbol{y}}_n)\right]d\boldsymbol{y}, \quad (11)$$

and $\underline{x}(\boldsymbol{y}) = \int_{\mathcal{Y}}k^*(\boldsymbol{y},\boldsymbol{y}')x(\boldsymbol{y}')d\boldsymbol{y}'$. Then we apply calculus of variations to the action integral, where the functional derivative of $S(x(\boldsymbol{y}),\underline{x}(\boldsymbol{y}))$ on the MAP estimator $\hat{x}(\boldsymbol{y})$ should be equal to zero: $\frac{\delta S}{\delta\hat{x}(\boldsymbol{y})}\delta\hat{x}(\boldsymbol{y}) + \frac{\delta S}{\delta\underline{\hat{x}}(\boldsymbol{y})}\delta\underline{\hat{x}}(\boldsymbol{y}) = 0$, which results in the exact MAP estimator,

$$\hat{x}(\boldsymbol{y}) = 2\sum_{n=1}^{N}h(\boldsymbol{y},\tilde{\boldsymbol{y}}_n)v_n, \quad v_n = \hat{x}(\tilde{\boldsymbol{y}}_n)^{-1}, \quad (12)$$

where $h(\boldsymbol{y},\boldsymbol{y}')$ is a transformed kernel function that solves a discretized Fredholm integral equation of the second kind [36],

$$h(\boldsymbol{y},\boldsymbol{y}') + 2\sum_{j=1}^{J}k(\boldsymbol{y},\boldsymbol{y}_j)\Delta_jh(\boldsymbol{y}_j,\boldsymbol{y}') = k(\boldsymbol{y},\boldsymbol{y}'). \quad (13)$$

See Appendix A for the detailed derivations. Equation (12) shows that the MAP estimator of SurvPP involves the representer theorem under the transformed kernel function $h(\boldsymbol{y},\boldsymbol{y}')$, and thus the Bayesian estimation reduces to a finite-dimensional optimization problem. Here, we call $h(\boldsymbol{y},\boldsymbol{y}')$ the equivalent kernel following studies by Flaxman et al. [12], Walder & Bishop [45], and Kim et al. [24]. Given the equivalent kernel, the unknown coefficients $v_n$ in (12) solve the simultaneous quadratic equations derived from (12),

$$r_n \triangleq 2\,v_n\sum_{n'=1}^{N}h(\tilde{\boldsymbol{y}}_n,\tilde{\boldsymbol{y}}_{n'})v_{n'} - 1 = 0, \quad n = 1,2,\ldots,N. \quad (14)$$

In this paper, we estimate a set of coefficients, $\{v_n\}_{n=1}^{N}$, by solving a minimization problem of the mean of the squared residuals, $\sum_{n=1}^{N}|r_n|^2/N$, with a popular gradient descent algorithm, *Adam* [26].

## 2.4 Equivalent Kernels

The equivalent kernel $h(\boldsymbol{y},\boldsymbol{y}')$ solves the discretized Fredholm integral equation (13). When the kernel function of GP has degenerate form with rank $M(<\infty)$ such that

$$k(\boldsymbol{y},\boldsymbol{y}') = \sum_{m=1}^{M}\phi_m(\boldsymbol{y})\phi_m(\boldsymbol{y}') = \boldsymbol{\phi}(\boldsymbol{y})^{\top}\boldsymbol{\phi}(\boldsymbol{y}'), \quad (15)$$

it is easily shown that the discretized Fredholm integral equation (13) can be solved analytically [2, 24] as,

$$h(\boldsymbol{y}, \boldsymbol{y}') = \boldsymbol{\phi}(\boldsymbol{y})^\top (\boldsymbol{I}_M + 2\boldsymbol{A})^{-1} \boldsymbol{\phi}(\boldsymbol{y}'), \quad \boldsymbol{A} = \sum_{j=1}^{J} \Delta_j \boldsymbol{\phi}(\boldsymbol{y}_j) \boldsymbol{\phi}(\boldsymbol{y}_j)^\top, \tag{16}$$

where $\boldsymbol{I}_M$ is the $M \times M$ identity matrix, and $\boldsymbol{\phi}(\boldsymbol{y}) = (\phi_1(\boldsymbol{y}), \phi_2(\boldsymbol{y}), \dots, \phi_M(\boldsymbol{y}))^\top$. Note that $\boldsymbol{A}$ is defined by a sum of outer products, which can be rewritten through a matrix-matrix multiplication as

$$\boldsymbol{A} = \boldsymbol{B}\boldsymbol{B}^\top, \quad \boldsymbol{B} = \left[ \sqrt{\Delta_1} \boldsymbol{\phi}(\boldsymbol{y}_1), \dots, \sqrt{\Delta_J} \boldsymbol{\phi}(\boldsymbol{y}_J) \right]. \tag{17}$$

Empirically, implementation by matrix-matrix multiplication (17) is substantially faster than the sum of outer products when $J \gg 1$.

When $k(\boldsymbol{y}, \boldsymbol{y}')$ has degenerate form with rank $M$, the relation (16) shows that the equivalent kernel $h(\boldsymbol{y}, \boldsymbol{y}')$ also has degenerate form obtained through Cholesky decomposition:

$$h(\boldsymbol{y}, \boldsymbol{y}') = (\boldsymbol{L}\boldsymbol{\phi}(\boldsymbol{y}))^\top (\boldsymbol{L}\boldsymbol{\phi}(\boldsymbol{y}')), \quad \boldsymbol{L}^\top \boldsymbol{L} = (\boldsymbol{I}_M + 2\boldsymbol{A})^{-1}. \tag{18}$$

The degenerate equivalent kernel (18) offers fast Bayesian estimation that scales linearly with $N$ (see Section 2.6). In this paper, we used the random feature map [37, 42, 43] of a Gaussian kernel to obtain a degenerate form of kernel ($M{=}500$).

If the covariate map, $\boldsymbol{y}_u(t)$, can be assumed to be smooth enough over time, a more accurate evaluation of $\boldsymbol{A}$ than equation (16) is possible (see Appendix D.1), but this was not exploited in the main experiments.

## 2.5 Predictive Distribution and Marginal Likelihood

One of the advantages of GP models over non-Bayesian approaches is that they can provide predictive distributions and marginal likelihoods, which enable us to perform uncertainty evaluations, hyper-parameter optimization, and model selection in Bayesian manner. Following the methodology with the path integral representation of GP [23, 24], SurvPP (1-2) adopts a Laplace approximation in the functional space, and finds the approximate form of the predictive distribution and the marginal likelihood. We only show the results due to space limitations. For details, see Appendix B.

The marginal likelihood, $p(\mathcal{D})$, is given as

$$\log p(\mathcal{D}) = \log |\boldsymbol{Z}| - \frac{1}{2} \log |\boldsymbol{I}_N + \boldsymbol{Z}^{-1} \boldsymbol{H}| - \frac{1}{2} \log |\boldsymbol{I}_M + 2\boldsymbol{A}|^{-1} + (\log 2 - 1)N, \tag{19}$$

where $\boldsymbol{I}_N$ is the identity matrix with size $N$, and

$$\boldsymbol{Z}_{nn'} = (2v_n^2)^{-1} \delta_{nn'}, \quad \boldsymbol{H}_{nn'} = h(\tilde{\boldsymbol{y}}_n, \tilde{\boldsymbol{y}}_{n'}), \quad \boldsymbol{h}(\boldsymbol{y}) = (h(\boldsymbol{y}, \tilde{\boldsymbol{y}}_1), \dots, h(\boldsymbol{y}, \tilde{\boldsymbol{y}}_N))^\top. \tag{20}$$

The predictive distribution of intensity function on a covariate value, $p(\lambda = x^2(\boldsymbol{y}))$, is given as

$$p_{\mu,\nu}(\lambda) = \frac{1}{\Gamma(\nu)\mu^\nu} \lambda^{\nu-1} \exp(-\lambda/\mu), \quad \mu = 2\sigma \frac{2\hat{x}^2 + \sigma}{\hat{x}^2 + \sigma}, \quad \nu = \frac{(\hat{x}^2 + \sigma)^2}{2\sigma(2\hat{x}^2 + \sigma)}, \tag{21}$$

where $\hat{x}$ is the abbreviation of $\hat{x}(\boldsymbol{y})$, and the predictive variance, $\sigma$, is defined by $\sigma = h(\boldsymbol{y}, \boldsymbol{y}) - \boldsymbol{h}(\boldsymbol{y})^\top (\boldsymbol{Z} + \boldsymbol{H})^{-1} \boldsymbol{h}(\boldsymbol{y})$. (21) represents the predictive distribution of the hazard function as a regression function of covariates. It is worth noting that the posterior process of SurvPP is given by a permanental process, and we can evaluate the distribution of the survival function and perform a risk-aware survival analysis by generating random samples of the estimated hazard function of covariates, $\lambda(\boldsymbol{y})$.

## 2.6 Computational Complexity

The computational complexity of evaluating the equivalent kernel (16) for covariate pair $(\boldsymbol{y}, \boldsymbol{y}')$ is $\mathcal{O}(M^3 + dJM + JM^2)$, where $\mathcal{O}(dJM)$ and $\mathcal{O}(JM^2)$ come from the computation of feature maps, $\phi_m(\boldsymbol{y}_j)$, and the sum of outer products (17), respectively.

When the equivalent kernel is given in degenerate form with rank $M$ ($< N$) such that $h(\boldsymbol{y}, \boldsymbol{y}') = \sum_{m=1}^{M} \varphi_m(\boldsymbol{y}) \varphi_m(\boldsymbol{y}')$, the objective function to be minimized in MAP estimation, $\sum_{n=1}^{N} |r_n|^2/N$,

incurs a linear computation with the number of observed events $N$, that is, $\mathcal{O}(NM)$, for each evaluation in gradient descent algorithms: The vector of residual, $\boldsymbol{r} = (r_1, \ldots, r_N)^\top$, can be expressed as $(\boldsymbol{R}(\boldsymbol{R}^\top \boldsymbol{v})) \odot (2\boldsymbol{v}) - \mathbf{1}$, where $\boldsymbol{R}$ is the $N \times M$ matrix defined by $\boldsymbol{R}_{nm} = \varphi_m(\boldsymbol{y}_n)$, and $\odot$ represents the Hadamard product. $\boldsymbol{R}$ costs $\mathcal{O}(dNM)$.

Given an equivalent kernel with degenerate form, the evaluation of the predictive variance, $\sigma = h(\boldsymbol{y}, \boldsymbol{y}) - \boldsymbol{h}(\boldsymbol{y})^\top (\boldsymbol{Z} + \boldsymbol{H})^{-1} \boldsymbol{h}(\boldsymbol{y})$, needs the computation of $\mathcal{O}(M^3 + NM^2)$: $N \times N$ matrix $\boldsymbol{H}$ can be decomposed into a product of $N \times M$ matrix $\boldsymbol{R}$ and its transpose as $\boldsymbol{H} = \boldsymbol{R}\boldsymbol{R}^\top$. The matrix inversion is transformed as $(\boldsymbol{Z} + \boldsymbol{R}\boldsymbol{R}^\top)^{-1} = \boldsymbol{Z}^{-1} - \boldsymbol{Z}^{-1}\boldsymbol{R}(\boldsymbol{I}_M + \boldsymbol{R}^\top \boldsymbol{Z}^{-1}\boldsymbol{R})^{-1}\boldsymbol{R}^\top \boldsymbol{Z}^{-1}$ through the Woodbury matrix identity, which costs $\mathcal{O}(M^3 + NM^2)$. Note that $\boldsymbol{Z}^{-1}\boldsymbol{R}$ and $\boldsymbol{R}^\top \boldsymbol{Z}^{-1}$ cost $\mathcal{O}(NM)$ because $\boldsymbol{Z}$ is a diagonal matrix.

In computing the marginal likelihood (19), the complexities of the first, the second, and the third terms are $\mathcal{O}(N)$, $\mathcal{O}(M^3 + NM^2)$, and $\mathcal{O}(M^3 + JM^2)$, respectively, where the matrix determinant lemma is used in computing the second term.

In total, the computational complexity of SurvPP is $\mathcal{O}(NMQ + (N + J)(d + M)M + M^3)$, where $Q$ is the number of gradient descent iterations. This feasible computation, which scales linearly with the number of observed events, $N$, the data size, $J$, and the covariate dimensionality, $d$, is achieved by exploiting the representer theorem. It should be emphasized here that data size $J$ is not part of the gradient descent iteration term. This is a clear advantage over conventional survival models which naively incur $\mathcal{O}((dN + dJ + JN)Q)$ computation: Typically, $J$ is larger than $N$ in a time-varying covariate scenario (see the counting process format in Section 2.2), and the discrepancy between $J$ and $N$ becomes more substantial when the measurements of covariates are made more frequently or/and over a longer period of time.

## 3 Related Work

**Survival Models for Time-invariant Covariates:** The most popular survival model is the Cox proportional hazards model [5] (CoxPH), which is a semi-parametric model that forms the hazard function as

$$\lambda(t) = h(t)\exp(\beta_1 y_1 + \beta_2 y_2 + \cdots), \tag{22}$$

where the base hazard function, $h(t)$, is obtained by the non-parametric Breslow/Fleming-Harrington estimator [3, 13], and the log-linear regression coefficient $(\beta_1, \ldots,)$ is estimated by maximizing the partial likelihood function. CoxPH is very efficient to compute and scales linearly with the number of events [39], but cannot address the nonlinear dependence of survival times on covariates. To overcome this limitation, a vast number of survival models have been proposed that replace the log-linear parametric function (22) with a non-linear one, such as generalized boosted models [38], random survival forests [17], Gaussian process models [11, 25], and deep neural network models [10, 22, 49]. For a comprehensive review, see Wang et al. [46].

**Survival Models for Time-varying Covariates:** The counting process format of input [1] plays a central role in extending static survival models into those suitable for time-varying covariates. For each individual, the counting process format splits her/his observation period into multiple short intervals, assigns a constant value of covariate to each interval, and marks the intervals as being censored when no event is present, resulting in a set of pseudo-individual right/left-censored observations with a constant covariate (see Section 2.2). CoxPH was extended to accommodate the counting process format, and the extended CoxPH can estimate the hazard function based on survival data with time-varying covariates [1]. To alleviate the simplest assumption in the extended CoxPH, generalized boosted models [16] and random survival forests [47, 48], both of which are ensemble approaches, have been also extended to accommodate the counting process format, where the extended CoxPH is employed as a weak learner or a tree. While they can address the nonlinear dependence of survival times on time-varying covariates, the large number of learners/trees needed for accurate predictions could make the algorithms too slow and ineffective for real-time applications. Also, Cygu et al. reported that random survival forests for time-varying covariates required substantial amounts of computer memory for large datasets [7].

**Exogeneity and Endogeneity of Covariates:** When considering survival analysis in the presence of time-varying covariates, we need to distinguish between exogenous and endogenous covariates. Kalbfleisch and Prentice [21] define an endogenous (internal) covariate as the output of a stochastic

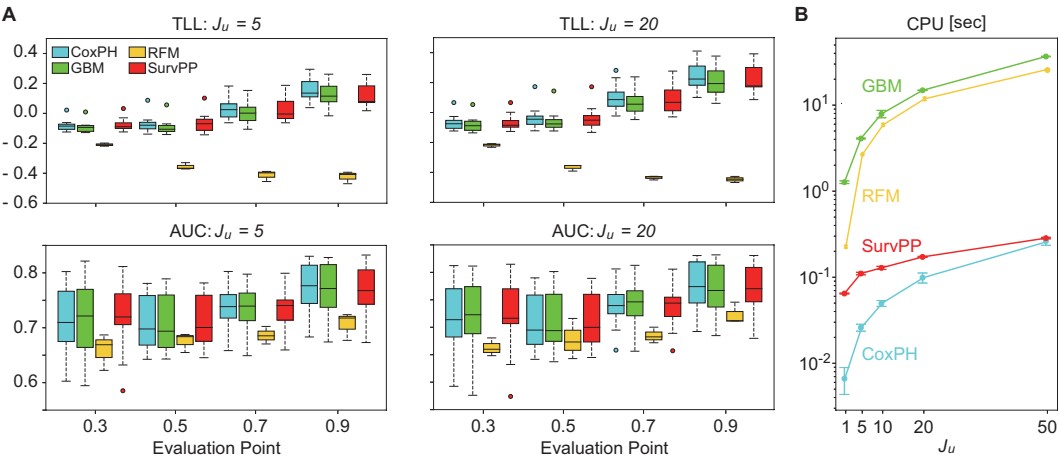

Figure 1: Performance on the dataset of log-linear hazard function $\lambda_{lin}(t)$. (A) Box plot of TLL and AUC as functions of evaluation point: the higher, the better. (B) The CPU times demanded for estimating a hazard function. The error bars represent the standard deviations. For GBM and SurvPP, the average cpu times over 9-points grid search of the hyperparameter are displayed.

process that is generated by the individual under study. In contrast, an exogenous (external) co-variate is not influenced by the individual under study. Diggle et al. [9] suggest using similar but slightly different definitions. The primary purpose of survival analysis with exogenous covariates is to estimate a hazard function as an explicit function of covariates, and to make predictions of failure times under various possible *future* covariate functions (i.e., what-if analysis). In contrast, the primary purpose of survival analysis with endogenous covariates is to make predictions of failure times by using the *past* observations of covariates, where joint modeling approaches with recurrent neural networks, which jointly model the stochastic process of covariates and failure times, have been developed intensively [29, 33]. In this paper, we consider exogenous covariates, and survival models for exogenous covariates cannot be compared directly with those for endogenous covariates because the tasks are different. Note that a joint modeling approach for exogenous time-varying covariates has been proposed very recently[32], which was not included in the benchmark models because it was published a month after this paper's submission.

**Permanental Process:** The permanental process is a variant of Gaussian Cox process that assumes the square root of hazard function to be generated from a Gaussian process [31]. Its computational advantages have recently been highlighted in machine learning research [12, 18, 24, 25, 30, 45]. In particular, the representer theorem in the permanental process has been exploited through the RKHS theory [12], the Mercer's theorem [45], and the path integral formulation [23, 24]. Although the key derivation of SurvPP is based on the path integral methodology used for augmented permanental process (APP) [24], SurvPP is a non-trivial extension of APP: (i) SurvPP can accommodate multiple trials of event sequence data, while APP assumes one trial of evet sequence; (ii) in SurvPP, the end time of observation is a stochastic variable that depends on the time of event occurrence, while APP assumes that the end time of observation is given. We discovered, for the first time, that the representer theorem holds for such a complicated point process or a survival model.

## 4 Experiments

We examined the validity of our proposed model by comparing it with conventional survival models on synthetic and real-world data. As benchmark models, we adopted Cox proportional hazards model (CoxPH), generalized boosted model (GBM), and random forest-based model (RFM): We implemented CoxPH and GBM with the established algorithms provided in the packages `survival.coxph` [41] and `gbm3.gbmt` [16], respectively; We implemented RFM by using class `ltrcrrf` in the open R code provided by Yao et al. [48]. The benchmark models can accommodate time-varying covariates via counting process format of input (`Surv(t0,t1,event)` in `survival`). For our proposal, we implemented SurvPP by using TensorFlow-2.10[1]. A MacBook Pro with 12-core CPU (Apple M2 Max) was used, where GPU was set as off (`tf.device('/cpu:0')`) for a

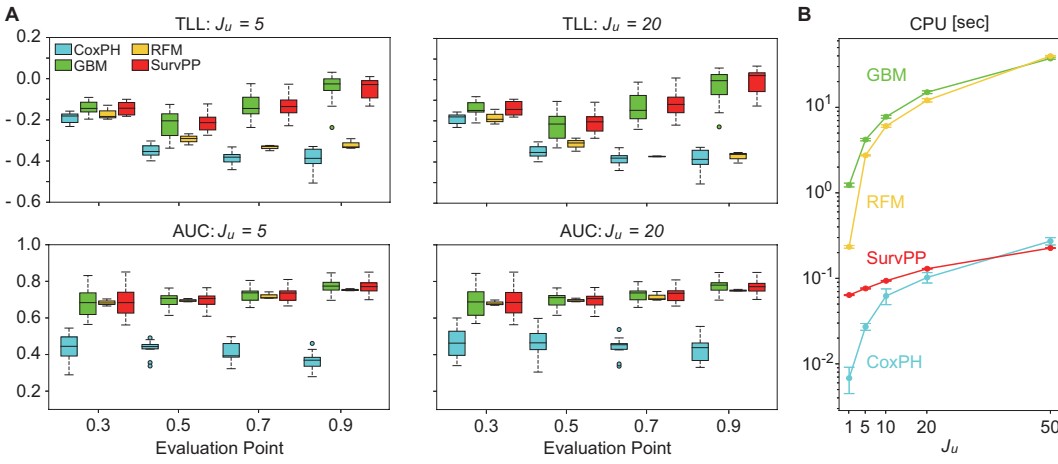

Figure 2: Performance on the dataset of non-linear hazard function $\lambda_{non}(t)$. (A) Box plot of TLL and AUC as functions of evaluation point: the higher, the better. (B) The CPU times demanded for estimating a hazard function. The error bars represent the standard deviations. For GBM and SurvPP, the average cpu times over 9-point grid search of the hyperparameter are displayed.

fair benchmark comparisons. For GBM and SurvPP, the hyper-parameters were optimized through 9-point grid search: the number of trees and the shrinkage for GBM, and the kernel parameters for SurvPP. For details of the model configurations, see Appendix C.

## 4.1 Synthetic Data

We created two survival datasets with time-varying 2-D covariate. One was generated from a hazard function with a log-linear dependence on covariates, and the other with a nonlinear dependence on covariates as follows:

$$\lambda_{lin}(t) = h(t) \exp\big[1.5y_1(t) + 3.5y_2(t)\big], \quad \lambda_{non}(t) = h(t) \exp\big[2 - 5(y_1^2(t) + y_2^2(t))\big], \quad (23)$$

where $h(t)$ is the base hazard function defined by a Weibull hazard function, $h(t) = 2 \cdot t^{3/2}$. For each of the datasets, we considered $U$ individuals, each of which had a 2-D covariate function of time:

$$\boldsymbol{y}_u(t) = \big(y_1^u(t), y_2^u(t)\big) = \big(\alpha_u \cos(2\pi\omega_u t + \pi\gamma_u), \alpha_u' \cos(2\pi\omega_u' t + \pi\gamma_u')\big), \quad u = 1, \ldots U, \quad (24)$$

where $\alpha_u(\alpha_u')$ and $\gamma_u(\gamma_u')$ were sampled uniformly over $[0,1]$, $\omega_u$ was sampled uniformly over $[5, 10]$, and $\omega_u'$ was sampled uniformly over $[20, 30]$. We set the censoring time as 1 for all individuals, and an event time generated from (23), denoted by $t_*$, was censored ($\eta_u = 0$) when $t_*$ was over 1: $T_u = \min(t_*, 1)$.

The predictive performances were evaluated based on the dynamic area under the ROC curve (AUC) [34] and the test log-likelihood (TLL),

$$\text{AUC}(t) = \frac{\sum_{u=1}^{U} \sum_{u'=1}^{U} \text{I}(T_u > t)\text{I}(T_{u'} \le t)w_u \text{I}(S(t|\boldsymbol{y}_u(\cdot)) \ge S(t|\boldsymbol{y}_{u'}(\cdot)))}{\big(\sum_{u=1}^{U} \text{I}(T_u > t)\big)\big(\sum_{u=1}^{U} \text{I}(T_u \le t)w_u\big)}, \quad (25)$$

$$\text{TLL}(t) = \frac{1}{U} \sum_{u=1}^{U} \Big[\text{I}(t \ge T_u) \cdot \eta_u \log \rho(T_u|\boldsymbol{y}_u(\cdot)) + \log S(\min(t, T_u)|\boldsymbol{y}_u(\cdot))\Big], \quad (26)$$

where $w_u$ is the inverse probability of censoring weight, and $\rho(t|\boldsymbol{y}_u(\cdot))$ and $S(t|\boldsymbol{y}_u(\cdot))$ are the estimated hazard and survival functions given covariate map $\boldsymbol{y}_u(\cdot)$, respectively. The dynamic AUC essentially estimates the C-index at each time, and is commonly used in the literature of survival analysis with time-varying covariates.

The evaluation point, $0 \le t_e \le 1$, was set as $t_e \in \{0.3, 0.5, 0.7, 0.9\}$. For each dataset, we randomly split the $U$ individuals into 10 subgroups, assigned one to test and the others to training data, and

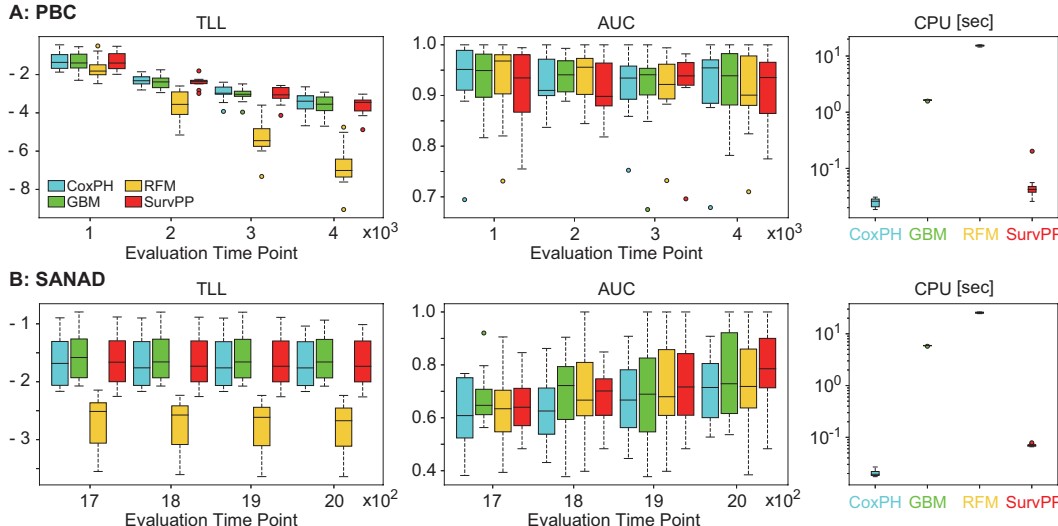

Figure 3: Performances on real-world datasets. TLL (the higher, the better), AUC (the higher, the better), and CPU times (the lower, the better) on PBC dataset (A) and SANAD dataset (B).

conducted 10-fold cross evaluation of the predictive performances. We reformatted the training data into counting process format by equally splitting individual $u$'s observation periods into $J_u$ sub-region and assuming the covariates to be constant within a sub-region; the survival models were applied to the training data. In this experiment, we considered $U = 10^3$ and $J_u \in \{1, 5, 10, 20, 50\}$: the number of observed events $N$ was 809 for $\lambda_{lin}$ and 818 for $\lambda_{non}$, respectively.

It should be noted here that when time-varying covariates are considered, the time duration from the entry, $t$, is itself a time-varying covariate. SurvPP considered a 3-D covariate map, $\bar{\boldsymbol{y}}_u(t) \equiv (y_1^u(t), y_2^u(t), t)$, and performed hazard function estimation, while the other models assumed the hazard function to be $h(t)f(\boldsymbol{y}(t))$, and $h(t)$ and $f(\boldsymbol{y})$ were estimated separately.

Figure 1 displays the predictive performance on the dataset of $\lambda_{lin}(t)$. Figure 1A shows that, except for RFM, there were no significant differences in the performances between the compared models, but CoxPH slightly outperformed the other models in TLL at some evaluation periods. The result is plausible because the underlying generative process was perfectly consistent with the log-linear and proportional assumptions of CoxPH. The comparison in TLL between $J_u = 5$ and $J_u = 20$ suggests that more frequently measured covariates would yield better estimations/predictions in scenarios of time-varying covariates. Therefore, the feasible computational complexity of SurvPP regarding $J_u$ is a clear advantage over the reference models, which is confirmed by Figure 1B.

Figure 2 displays the predictive performance on the dataset of $\lambda_{non}(t)$, showing that the non-parametric models, SurvPP and GBM, significantly outperformed CoxPH, while the performance gaps between SurvPP and GBM were marginal and of no significance. However, Figure 2B shows that SurvPP could estimate the hazard function hundreds of times faster than GBM when covariates were measured frequently ($J_u = 50$), and was even faster than the simplest CoxPH. The preferable scalability of SurvPP's algorithm regarding $J_u$ comes from the fact that $J_u$ (or $J = \sum_u J_u$) is not part of the iterative optimization (Section 2.6), which is a fruit of SurvPP's representer theorem. RFM was comparable with SurvPP and GBM in AUC, but was worse than either of them in TLL. RFM might need more careful tuning of the hyper-parameters, but this was not fully investigated this time.

We conducted additional experiments on independent validation datasets (see Appendix D.2), and on larger synthetic datasets ($U \leq 10^5$) to examine the model's computation scalability regarding the event number $N$ (see Appendix D.3).

## 4.2 Real-world Data

We examined the validity of SurvPP against the benchmark models on two real-world survival data sets, *Mayo Clinic Primary Biliary Cholangitis data* (PBC) and *Standard And New Antiepileptic*

*Drugs study data* (SANAD), provided by R packages `survival` (LGPL-3) [41] and `joineR` (GPL-3) [35], respectively. In PBC, 312 patients with primary biliary cirrhosis were enrolled in a randomized medical trial at the Mayo Clinic between 1974 and 1984 [8], where events were the time of death; one static and eleven time-varying covariates were measured at entry and at yearly intervals, which include age at entry, alkaline phosphotase, logarithm of serum albumin, presence of ascites, aspartate aminotransferase, logarithm of serum bilirubin, serum cholesterol, condition of edema, presence of hepatomegaly or enlarged liver, platelet count, logarithm of prothrombin time, and presence or absence of spiders. SANAD was an unblind randomized trial that recruited patients with epilepsy for whom carbamazepine (CBZ) was considered to be standard treatment and they were randomized to CBZ or the newer drug lamotrigine (LTG), where 605 patients were included and event was the time to treatment failure; we adopted calibrated dose as a time-varying covariate, and three static covariates including age of patient at randomization, gender, and randomized treatment (CBZ or LTG); calibrated dose was measured at 166-day intervals on average, and we used a linear interpolation to obtain the values of calibrated dose at which the measurement intervals were regularly trisected. We randomly split the individuals into 10 subgroups, assigned one to test and the others to training data, and conducted 10-fold cross evaluation of the predictive performances.

Figure 3A displays the predictive performance on PBC. It shows that CoxPH achieved very high AUCs (> 0.9), suggesting that the underlying generative process could be consistent with CoxPH's simple assumption of log-linearity and proportionality. Thus the semi-parametric model, CoxPH, is likely to achieve equal or slightly better performances than the nonparametric models, which is consistent with the result. Figure 3B plots the predictive performance on SANAD, where the not so high AUCs of CoxPH suggests that the underlying dependence of intensity on covariates is nonlinear. The figure shows that SurvPP achieved better performance than CoxPH, and achieved comparable performance while being substantially faster than GBM.

# 5  Conclusions

We have proposed a non-parametric Bayesian survival model to address survival data with time-varying covariates. We tailored a permanental process such that the latent hazard function is defined on covariate space and right-censored observations in a counting process format can be handled, which we call the survival permanental process (SurvPP). Through the path integral formulation of Gaussian process, we showed that SurvPP encompasses a representer theorem, and derived a feasible estimation algorithm that scales linearly with the number of observed events. We evaluated SurvPP on synthetic data, confirming that it achieved comparable predictive accuracy while being tens to hundreds of times faster than state-of-the-art methods.

**Limitations & future work:** We examined SurvPP for relatively low-dimensional covariates, and its potential suitability for high-dimensional covariates remains to be clarified. Because SurvPP is based on a normal Gaussian process, the (equivalent) kernel function might be too simple to discover meaningful representations in high-dimensional covariate data. A promising direction is to apply deep kernel learning to SurvPP, where high-dimensional covariates are transformed by nonlinear mapping with a deep architecture. A technical issue is that we could not search an appropriate set of (a lot of) parameters of neural networks, because our proposed scheme performs the hyper-parameter optimization by grid search, not by gradient descent. Also, Gaussian kernels, which were used in the paper, naively require a scale parameter for each dimension of data, resulting in a high dimensional kernel parameter for high-dimensional covaritate scenarios. The grid search becomes prohibitively costly with high dimensional parameters, but we addressed the problem by a well-known approach that normalizes (e.g., centering and scaling) each dimension of the data and puts a common scale parameter across all dimensions of normalized data. This approach empirically works robustly, but a more sophisticated approach could improve the performance of SurvPP. Variational Bayesian approximations with inducing points might address the technical issues of hyper-parameter optimization, and thus is an important next step in our study.

As in ordinary permanental processes, the nodal line problem could arise in SurvPP: the posterior distribution of the latent variable $x(\cdot)$ has many local modes since $\pm x(\cdot)$ can lead to similar hazard functions $\lambda = x^2(\cdot)$, and artificial zero crossings of $x(\cdot)$ could happen, especially on locations where the hazard function is low. John and Hensman [18] have proposed to extend the quadratic link function to include an offset parameter $\beta$, so that $\lambda(\boldsymbol{y}) = (x(\boldsymbol{y}) + \beta)^2$, which is valid for SurvPP.

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

## A Derivation of MAP Estimator

We detail the derivation of the MAP estimator (12). The functional derivative of $S(x(\boldsymbol{y}), \underline{x}(\boldsymbol{y}))$ should be zero on MAP estimator $\hat{x}(\boldsymbol{y})$:

$$
\begin{aligned}
\delta S\big(\hat{x}(\boldsymbol{y}), \underline{\hat{x}}(\boldsymbol{y})\big) &= \int_{\mathcal{Y}} \left[ \frac{\delta S}{\delta \hat{x}(\boldsymbol{y})} \delta x(\boldsymbol{y}) + \frac{\delta S}{\delta \underline{\hat{x}}(\boldsymbol{y})} \delta \underline{x}(\boldsymbol{y}) \right] d\boldsymbol{y} + O((\delta x)^2) \\
&\simeq \int_{\mathcal{Y}} \left[ 2 \sum_{j=1}^{J} \Delta_j \hat{x}(\boldsymbol{y}) \delta(\boldsymbol{y} - \boldsymbol{y}_j) - \sum_{n=1}^{N} \frac{2}{\hat{x}(\boldsymbol{y})} \delta(\boldsymbol{y} - \tilde{\boldsymbol{y}}_n) + \frac{1}{2} \underline{\hat{x}}(\boldsymbol{y}) \right] \delta x d\boldsymbol{y} \\
&\qquad\qquad\qquad\qquad\qquad\qquad\qquad\qquad\qquad\qquad + \int_{\mathcal{Y}} \frac{1}{2} \hat{x}(\boldsymbol{y}) \delta \underline{x} d\boldsymbol{y} \\
&= \int_{\mathcal{Y}} \left[ 2 \sum_{j=1}^{J} \Delta_j \hat{x}(\boldsymbol{y}) \delta(\boldsymbol{y} - \boldsymbol{y}_j) - \sum_{n=1}^{N} \frac{2}{\hat{x}(\boldsymbol{y})} \delta(\boldsymbol{y} - \tilde{\boldsymbol{y}}_n) + \underline{\hat{x}}(\boldsymbol{y}) \right] \delta x d\boldsymbol{y} = 0,
\end{aligned}
$$

where the following relation was used,

$$
\begin{aligned}
\int_{\mathcal{Y}} \hat{x}(\boldsymbol{y}) \delta \underline{x} d\boldsymbol{y} &= \int_{\mathcal{Y}} \hat{x}(\boldsymbol{y}) \int_{\mathcal{Y}} k^*(\boldsymbol{y}, \boldsymbol{y}') \delta x(\boldsymbol{y}') d\boldsymbol{y}' d\boldsymbol{y} \\
&= \int_{\mathcal{Y}} d\boldsymbol{y}' \delta x(\boldsymbol{y}') \int_{\mathcal{Y}} k^*(\boldsymbol{y}, \boldsymbol{y}') \hat{x}(\boldsymbol{y}) d\boldsymbol{y} \\
&= \int_{\mathcal{Y}} \underline{\hat{x}}(\boldsymbol{y}') \delta x d\boldsymbol{y}'. \quad \because) \; k^*(\boldsymbol{y}, \boldsymbol{y}') = k^*(\boldsymbol{y}', \boldsymbol{y})
\end{aligned}
$$

Thus the following equation is derived,

$$
\underline{\hat{x}}(\boldsymbol{y}) + 2 \sum_{j=1}^{J} \Delta_j \hat{x}(\boldsymbol{y}_j) \delta(\boldsymbol{y} - \boldsymbol{y}_j) = \sum_{n=1}^{N} \frac{2}{\hat{x}(\tilde{\boldsymbol{y}}_n)} \delta(\boldsymbol{y} - \tilde{\boldsymbol{y}}_n), \quad \boldsymbol{y} \in \mathcal{Y}. \tag{A1}
$$

By applying operator $\mathcal{K}$ to (A1), we obtain a linear integral equation that derives the MAP estimator $\hat{x}(\boldsymbol{y})$ as follows,

$$
\hat{x}(\boldsymbol{y}) + 2 \sum_{j=1}^{J} \Delta_j \hat{x}(\boldsymbol{y}_j) k(\boldsymbol{y}, \boldsymbol{y}_j) = 2 \sum_{n=1}^{N} k(\boldsymbol{y}, \tilde{\boldsymbol{y}}_n) \hat{x}(\tilde{\boldsymbol{y}}_n)^{-1}, \quad \boldsymbol{y} \in \mathcal{Y}. \tag{A2}
$$

The linearity of the integral equation permits a representation of the form

$$
\hat{x}(\boldsymbol{y}) = 2 \sum_{n=1}^{N} h(\boldsymbol{y}, \tilde{\boldsymbol{y}}_n) \hat{x}(\tilde{\boldsymbol{y}}_n)^{-1},
$$

where $h(\boldsymbol{y}, \boldsymbol{y}')$ is a positive semi-definite kernel that solves integral equation (13). Note that the derivations follow Kim [23].

## B Derivation of Predictive Distribution and Marginal Likelihood

We now know the mode of the posterior, $\hat{x}(\boldsymbol{y})$, and consider a Taylor expansion of functional action potential $S(x(\boldsymbol{y}), \underline{x}(\boldsymbol{y}))$ centered on the mode such that

$$
S\big(x(\boldsymbol{y}), \underline{x}(\boldsymbol{y})\big) \simeq S\big(\hat{x}(\boldsymbol{y}), \underline{\hat{x}}(\boldsymbol{y})\big) + \frac{1}{2} \iint_{\mathcal{Y} \times \mathcal{Y}} \sigma^*(\boldsymbol{y}, \boldsymbol{y}')(x(\boldsymbol{y}) - \hat{x}(\boldsymbol{y}))(x(\boldsymbol{y}') - \hat{x}(\boldsymbol{y}')) d\boldsymbol{y} d\boldsymbol{y}', \tag{B1}
$$

where $\sigma^*(\boldsymbol{y}, \boldsymbol{y}') = \frac{\delta^2 S(x,x)}{\delta x(\boldsymbol{y}) \delta x(\boldsymbol{y}')}\big|_{x=\hat{x}}$ is the second derivative of $S$. The first term in the Taylor expansion vanishes due to the stationary condition. The quadratic approximation of the action integral corresponds to the approximation of the posterior process by a GP, and the predictive covariance or the kernel function for the posterior GP, denoted by $\sigma(\boldsymbol{y}, \boldsymbol{y}')$, can be obtained by the functional inversion of $\sigma^*(\boldsymbol{y}, \boldsymbol{y}')$, which results in

$$
\sigma(\boldsymbol{y}, \boldsymbol{y}') = h(\boldsymbol{y}, \boldsymbol{y}') - \boldsymbol{h}(\boldsymbol{y})^{\top}(\boldsymbol{Z} + \boldsymbol{H})^{-1} \boldsymbol{h}(\boldsymbol{y}'), \tag{B2}
$$

where the definitions of $\boldsymbol{Z}$ and $\boldsymbol{H}$ are taken from (20). The full derivation of (B2) is given in [24]. When the latent function $x(\boldsymbol{y})$ follows a posterior GP with mean of $\hat{x}(\boldsymbol{y})$ and kernel $\sigma(\boldsymbol{y}, \boldsymbol{y}')$, it is easily verified that the value of the squared function, $\lambda = x^2(\boldsymbol{y})$, on each point of the covariate domain $\boldsymbol{y} \in \mathcal{Y}$ follows a Gamma distribution defined by (21).

Furthermore, under Laplace approximation (B1), we can obtain the marginal likelihood, $p(\mathcal{D})$, in (10) by performing the path integral as,

$$\log p(\mathcal{D}) = \log \int \exp\left[-S\big(x(\boldsymbol{y}), \underline{x}(\boldsymbol{y})\big) - \frac{1}{2}\log|\mathcal{K}|\right]\mathscr{D}x \simeq -S\big(\hat{x}(\boldsymbol{y}), \hat{\underline{x}}(\boldsymbol{y})\big) + \frac{1}{2}\log\frac{|\Sigma|}{|\mathcal{K}|}, \text{ (B3)}$$

where $|\Sigma|$ is the functional determinant of integral operator $\Sigma = \int_{\mathcal{Y}} \cdot\, \sigma(\boldsymbol{y}, \boldsymbol{y}')d\boldsymbol{y}'$. We can rewrite the result in a more tractable form by substituting (11, B2) into (B3):

$$\log p(\mathcal{D}) = \log|\boldsymbol{Z}| - \frac{1}{2}\log|\boldsymbol{I}_N + \boldsymbol{Z}^{-1}\boldsymbol{H}| - \frac{1}{2}\log|\boldsymbol{I}_M + 2\boldsymbol{A}|^{-1} + (\log 2 - 1)N.$$

Full derivations are provided in [24].

# C  Model Configuration

## C.1  Synthetic Data

### Survival Permanental Process (SurvPP)

We set the number of features for Random feature map [37] ($M$), learning parameter ($lr$), and stop condition ($G$) for Adam [26] as follows:

$$M = 500, \quad lr = 0.05, \quad G < 10^{-5}.$$

We applied to SurvPP a multiplicative Gaussian kernel

$$k(\boldsymbol{y}, \boldsymbol{y}') = \prod_{d=1}^{3} e^{-(\theta(y_d - y_d'))^2}, \quad \boldsymbol{y} = (t, y_1, y_2),$$

where hyper-parameter $\theta$ was optimized for each data by maximizing the marginal likelihood through grid search. In the experiments on synthetic data, we selected a set of nine values for $\theta$ as the grid points,

$$\theta \in \{0.1,\ 0.2,\ 0.5,\ 0.7,\ 1.0,\ 2.0,\ 5.0,\ 7.0,\ 10.0\}.$$

We implemented SurvPP by using TensorFlow-2.10. A MacBook Pro with 12-core CPU (Apple M2 Max) was used, with the GPU inactivated (`tf.device('/cpu:0')`) for a fair comparison with the benchmarks.

### Cox Proportional Hazards Model (CoxPH)

We implemented CoxPH through package `survival.coxph` (LGPL-3) [41]. The calls in `coxph` to fit a model and compute a base hazard function were

$$> \texttt{cfit} = \texttt{coxph}(\texttt{Surv}(\texttt{Start}, \texttt{Stop}, \texttt{Event}) \sim \texttt{cov1} + \texttt{cov2}, \texttt{df})$$
$$> \texttt{sfit} = \texttt{survfit}(\texttt{cfit}, \texttt{list}(\texttt{cov1} = 0, \texttt{cov2} = 0)),$$

where `df` was the survival data in counting process format.

### Generalized Boosted Model (GBM)

We implemented GBM through package `gbm3.gbmt` [16] (GPL). The call in `gbmt` to fit a model was

$$> \texttt{gfit} = \texttt{gbmt}(\texttt{Surv}(\texttt{Start}, \texttt{Stop}, \texttt{Event}) \sim \texttt{cov1} + \texttt{cov2}, \texttt{data} = \texttt{df},$$
$$\texttt{distribution} = \texttt{gbm\_dist}(\text{``CoxPH''}),$$
$$\texttt{cv\_folds} = 10, \texttt{train\_params} = \texttt{params},$$
$$\texttt{par\_details} = \texttt{gbmParallel}(\texttt{num\_threads} = 12)),$$

where `params` represents the hyperparameter. We selected a set of nine hyperparameters for the grid search,

$$\texttt{num\_trees} \in \{500, 1000, 2000\} \ \times \ \texttt{shrinkage} \in \{0.001, 0.005, 0.01\},$$

and found the one that minimized the cross validation error (`gfit$valid.error`), where `num_trees` and `shrinkage` represent the number of trees and the shrinkage/learning rate, respectively.

### *Random Forest-based Model (RFM)*

We implemented RFM through package `LTRCforests` (GPL) [48]. The call to fit a model was

$$> \texttt{rfit} = \texttt{ltrcrrf}(\texttt{Surv}(\texttt{Start}, \texttt{Stop}, \texttt{Event}) \sim \texttt{cov1} + \texttt{cov2}, \texttt{data} = \texttt{df},$$
$$\texttt{id} = \texttt{ID}, \texttt{mtry} = \texttt{ceiling}(10), \texttt{ntree} = 100).$$

## C.2 Real-world Data: PBC

### *Survival Permanental Process (SurvPP)*

We set the number of features for Random feature map [37] ($M$), learning parameter ($lr$), and stop condition ($G$) for Adam [26] as follows:

$$M = 500, \quad lr = 50, \quad G < 10^{-5}.$$

We applied to SurvPP a multiplicative Gaussian kernel

$$k(\boldsymbol{y}, \boldsymbol{y}') = \prod_{d=1}^{13} e^{-(\theta(y_d - y'_d))^2}, \quad \boldsymbol{y} = (t, y_1, \ldots, y_{12}),$$

where the hyper-parameter $\theta$ was optimized for each data by maximizing the marginal likelihood through grid search. In the experiments on synthetic data, we selected a set of nine values for $\theta$ as the grid points,

$$\theta \in \{0.01, \ 0.02, \ 0.03, \ 0.04, \ 0.05, \ 0.06, \ 0.07, \ 0.08, \ 0.09\}.$$

Here, we normalized the 13 covariates so that $y_d \rightarrow 0.1(y_d - \text{mean}[y_d])/\text{std}[y_d]$.

As in the experiments on synthetic data, GPU was set off (`tf.device('/cpu:0')`) for a fair comparison with the benchmarks.

### *Cox Proportional Hazards Model (CoxPH)*

The calls in `coxph` to fit a model and compute a base hazard function were

$$> \texttt{cfit} = \texttt{coxph}(\texttt{Surv}(\texttt{Start}, \texttt{Stop}, \texttt{Event}) \sim \texttt{age} + \texttt{edema} + \texttt{alk.phos} + \texttt{chol} + \texttt{ast}$$
$$+ \texttt{platelet} + \texttt{spiders} + \texttt{hepato} + \texttt{ascites} + \texttt{albumin} + \texttt{bili} + \texttt{protime}, \texttt{df})$$
$$> \texttt{sfit} = \texttt{survfit}(\texttt{cfit}, \texttt{list}(\texttt{age} = 0, \texttt{edema} = 0, \texttt{alk.phos} = 0, \texttt{chol} = 0, \texttt{ast} = 0,$$
$$\texttt{platelet} = 0, \texttt{spiders} = 0, \texttt{hepato} = 0, \texttt{ascites} = 0, \texttt{albumin} = 0,$$
$$\texttt{bili} = 0, \texttt{protime} = 0)),$$

where `df` was the survival data in counting process format.

### *Generalized Boosted Model (GBM)*

The call in `gbmt` to fit a model was

$$> \texttt{gfit} = \texttt{gbmt}(\texttt{Surv}(\texttt{Start}, \texttt{Stop}, \texttt{Event}) \sim \texttt{age} + \texttt{edema} + \texttt{alk.phos} + \texttt{chol} + \texttt{ast}$$
$$+ \texttt{platelet} + \texttt{spiders} + \texttt{hepato} + \texttt{ascites} + \texttt{albumin} + \texttt{bili}$$
$$+ \texttt{protime}, \texttt{data} = \texttt{df},$$
$$\texttt{distribution} = \texttt{gbm\_dist}(\text{``CoxPH''}),$$
$$\texttt{cv\_folds} = 10, \texttt{train\_params} = \texttt{params},$$
$$\texttt{par\_details} = \texttt{gbmParallel}(\texttt{num\_threads} = 12)),$$

where `params` represents the hyperparameter. We selected a set of nine hyperparameters for the grid search,

$$\texttt{num\_trees} \in \{500, 1000, 2000\} \times \texttt{shrinkage} \in \{0.001, 0.005, 0.01\},$$

and found the one that minimized the cross validation error (`gfit$valid.error`), where `num_trees` and `shrinkage` represent the number of trees and the shrinkage/learning rate, respectively.

### *Random Forest-based Model (RFM)*

The call to fit a model was

```
> rfit = ltrcrrf(Surv(Start, Stop, Event) ∼ age + edema + alk.phos + chol + ast
                 +platelet + spiders + hepato + ascites + albumin + bili
                 +protime, data = df, id = ID, stepFactor = 1.5).
```

## C.3  Real-world Data: SANAD

### *Survival Permanental Process (SurvPP)*

We set the number of features for Random feature map [37] ($M$), learning parameter ($lr$), and stop condition ($G$) for Adam [26] as follows:

$$M = 500, \quad lr = 10, \quad G < 10^{-5},$$

We applied to SurvPP a multiplicative Gaussian kernel

$$k(\boldsymbol{y}, \boldsymbol{y}') = 0.1 \prod_{d=1}^{5} e^{-(\theta(y_d - y_d'))^2}, \quad \boldsymbol{y} = (t, y_1, \ldots, y_4),$$

where hyper-parameter $\theta$ was optimized for each data by maximizing the marginal likelihood through grid search. In the experiments on synthetic data, we selected a set of nine values for $\theta$ as the grid points,

$$\theta \in \{0.6, \ 0.7, \ 0.8, \ 0.9, \ 1.0, \ 1.1, \ 1.2, \ 1.3, \ 1.4\}.$$

Here, we normalized the 5 covariates so that $y_d \to 0.1(y_d - \text{mean}[y_d])/\text{std}[y_d]$.

As in the experiments on synthetic data, GPU was set off (`tf.device('/cpu:0')`) for a fair comparison with the benchmarks.

### *Cox Proportional Hazards Model (CoxPH)*

The calls in `coxph` to fit a model and compute a base hazard function were

```
> cfit = coxph(Surv(Start, Stop, Event) ∼ age + gender + treat + dose, df)
> sfit = survfit(cfit, list(age = 0, gender = 0, treat = 0, dose = 0)),
```

where `df` was the survival data in counting process format.

### *Generalized Boosted Model (GBM)*

The call in `gbmt` to fit a model was

```
> gfit = gbmt(Surv(Start, Stop, Event) ∼ age + gender + treat + dose,
             data = df, distribution = gbm_dist("CoxPH"),
             cv_folds = 10, train_params = params,
             par_details = gbmParallel(num_threads = 12)),
```

where `params` represents the hyperparameter. We selected a set of nine hyperparameters for the grid search,

$$\texttt{num\_trees} \in \{500, 1000, 2000\} \times \texttt{shrinkage} \in \{0.001, 0.005, 0.01\},$$

and found the one that minimized the cross validation error (`gfit$valid.error`), where `num_trees` and `shrinkage` represent the number of trees and the shrinkage/learning rate, respectively.

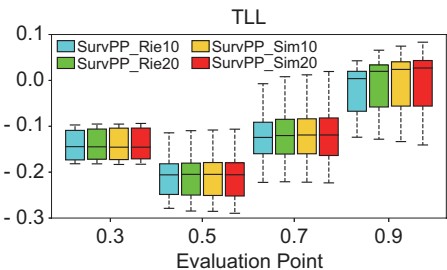 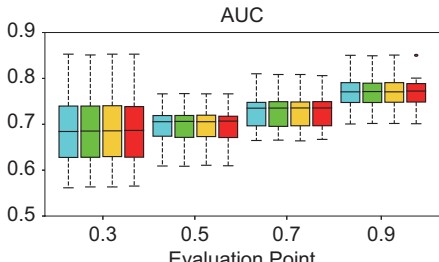

Figure D1: Performances on the dataset of non-linear hazard function $\lambda_{non}(t)$ for different numerical integration rules. SurvPP_Rie# and SurvPP_Sim# represent SurvPPs with Riemann sum and Simpson's rule approximations of integration, respectively. # is the number of representative time points for covariate, $J_u$.

*Random Forest-based Model (RFM)*

The call to fit a model was

```
> rfit = ltrcrrf(Surv(Start, Stop, Event) ~ age + gender + treat + dose,
                 data = df, id = ID, stepFactor = 1.5).
```

# D    Additional Experiments

## D.1    More Accurate Evaluation of Equivalent Kernel

The functional form of covariate map, $\boldsymbol{y}(t)$, involves the sum of outer products of feature maps, $\boldsymbol{A}$, defined by Equation (16), and $\boldsymbol{A}$ can be considered as a Riemann sum approximation of the time integral as follows:

$$\boldsymbol{A} = \sum_{u=1}^{U} \int_{0}^{T_u} \boldsymbol{\phi}(\boldsymbol{y}_u(t))\boldsymbol{\phi}(\boldsymbol{y}_u(t))^\top dt \sim \sum_{j=1}^{J} \Delta_j \boldsymbol{\phi}(\boldsymbol{y}_j)\boldsymbol{\phi}(\boldsymbol{y}_j)^\top. \tag{D1}$$

Thus, when the covariate map, $\boldsymbol{y}_u(t)$, and the feature map, $\boldsymbol{\phi}(\boldsymbol{y})$, are both smooth, the Riemann sum approximation can be replaced by a more accurate approximation,

$$\boldsymbol{A} = \sum_{u=1}^{U} \int_{0}^{T_u} \boldsymbol{\phi}(\boldsymbol{y}_u(t))\boldsymbol{\phi}(\boldsymbol{y}_u(t))^\top dt \sim \sum_{j=1}^{J} w_j \boldsymbol{\phi}(\boldsymbol{y}_j)\boldsymbol{\phi}(\boldsymbol{y}_j)^\top, \tag{D2}$$

where the weights, $w_j$, depend on the approximation rule. This time, we adopt Simpson's 1/3 rule, and add an experiment on the synthetic nonlinear data, $\lambda_{non}(t)$, to check for an improvement in accuracy. Note that Simpson's 1/3 rule corresponds to quadratic interpolation. Figure D1 shows that adopting Simpson's rule achieved better predictive performance on TLL with a smaller discretization number ($J_u$).

## D.2    Experiment on Independent Validation Data Sets

In Section 4, we evaluated the performances on synthetic data with the cross-validation approach because it is common in the machine learning literature. But synthetic data offers the unique advantage of being able to generate as much data as needed, and we here conducted the performance assessment with the independent validation approach. Figure D2 displays the result on independent validation sets of synthetic data ($U = 900$ for each of 10 training data, and $U = 100$ for each of 10 test data), showing the result similar to the cross-validation approach (Figure 1-2).

## D.3    Experiment on Larger Data Sets

To examine the computation scalability of the compared models versus the number of observed events $N$, we created data sets with user size $U \in \{10^3, 10^4, 5 \cdot 10^4, 10^5\}$ according to the nonlinear

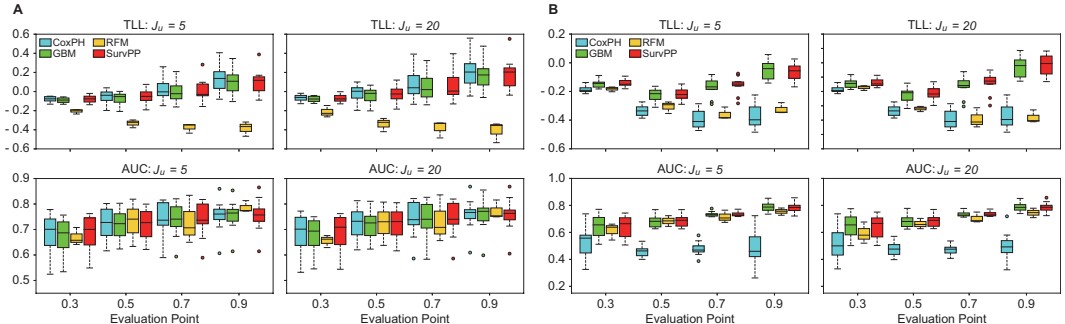

Figure D2: Performance on the independent validation synthetic datasets. (A) Log-linear hazard function $\lambda_{lin}(t)$. (B) Non-linear hazard function $\lambda_{non}(t)$. The results are consistent with Figure 1-2

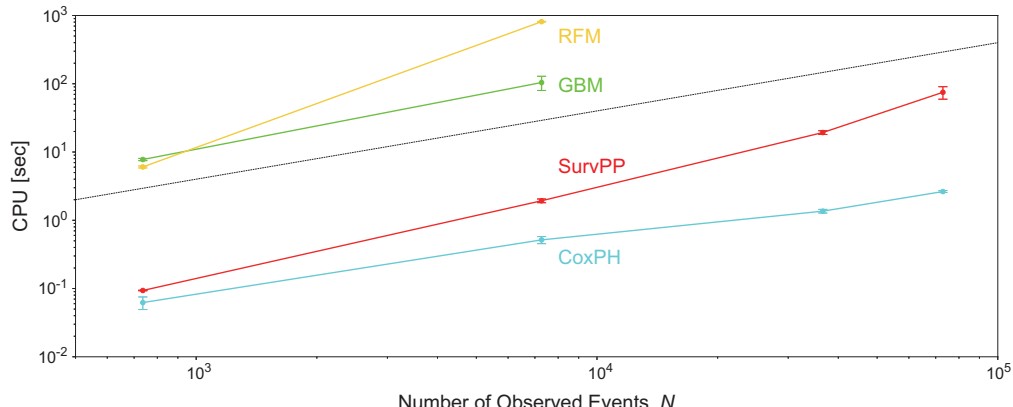

Figure D3: The CPU times demanded for estimating a hazard function versus the number of observed events. The error bars represent the standard deviations across 10 trials. The dashed line represents a line of CPU $\propto N$ as reference. For GBM and SurvPP, the average cpu times over 9-point grid search of the hyperparameter are displayed. The CPU times of GBM and RFM exceeded $10^3$ seconds with $N > 10^4$, and the estimations were given up.

scenario (see Section 4.1),

$$\lambda_{non}(t) = h(t) \exp\left[2 - 5(y_1^2(t) + y_2^2(t))\right], \quad h(t) = 2 \cdot t^{3/2},$$

which resulted in the data sets with $N \in \{818, 8082, 40660, 81066\}$. Here, we set $J_u$ to 10 for the counting process format of data. For each dataset, we randomly split the $U$ individuals into 10 subgroups, repeated assigning 9 subgroups to training data, and conducted 10 trials of evaluations of the CPU times demanded for estimating a hazard function. Figure D3 displays the CPU times as function of $N$ of training data. It shows that the CoxPH computation clearly scaled linearly with $N$, while that of SurvPP seems to be a little more than linear. This is because that each iteration of gradient descent algorithm scaled linearly with $N$, but the number of iterations to meet stop condition $G < 10^{-5}$ increased moderately with $N$. Among the non-parametric approaches (SurvPP, GBM, and RFM), SurvPP achieved the fastest computation at hundreds of times faster than the others, regardless of the number of observed events $N$.

## D.4 Sensitivity to Hyper-Parameters

Kernel methods/GP models are generally sensitive to kernel parameter values, and a effective way of optimizing the kernel parameter is essential. Figure D4 plots AUC performance on PBC dataset for SurvPPs with different search ranges of kernel parameters: the range of SurvPP$_0$ is $\theta \in \{0.1, 0.2, 0.5, 0.7, 1.0, 2.0, 5.0, 7.0, 10.0\}$, which was used in the supplement; the range of SurvPP$_1$ is $\theta \in \{0.01, 0.02, 0.03, \ldots, 0.09\}$. Figure D4 shows that SurvPP$_1$ achieved substantially

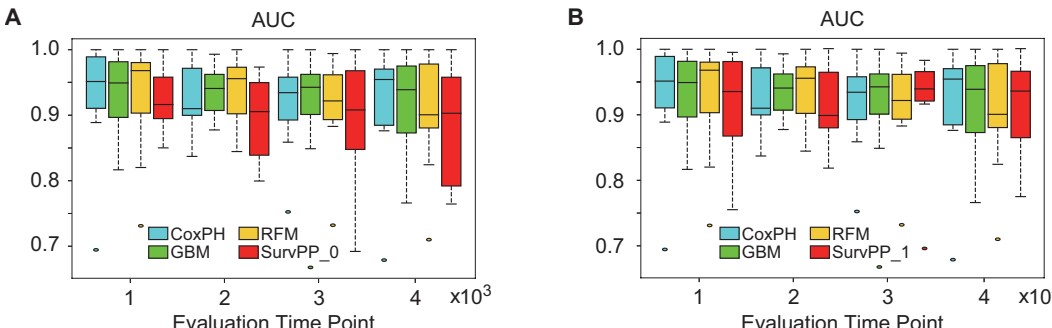

Figure D4: Performances on PBC dataset for SurvPPs with different search ranges of kernel parameters. (A) The search range of $SurvPP_0$ is $\theta \in \{0.1, 0.2, 0.5, 0.7, 1.0, 2.0, 5.0, 7.0, 10.0\}$. (B) The search range of $SurvPP_1$ is $\theta \in \{0.01, 0.02, 0.03, \ldots, 0.09\}$.

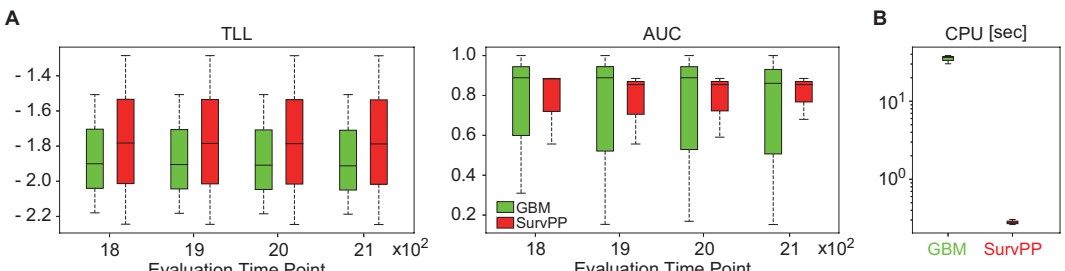

Figure D5: Performances on MIMIC III dataset. (A) Box plot of TLL and AUC as functions of evaluation point: the higher, the better. (B) The CPU times demanded for estimating a hazard function. The error bars represent the standard deviations. The average cpu times over 9-point grid search of the hyperparameter are displayed.

better AUC performance than $SurvPP_0$, where $SurvPP_1$ is displayed in Figure 3. A more sophisticated algorithm for hyperparameter optimization could enhance the performance of SurvPP, which is the next step in our study.

### D.5 Experiment on ICU Data Set

We examined the validity of SurvPP against GBM on *MIMIC-III Clinical Database* (MIMIC III), the large publicly available dataset of over 50,000 ICU admissions from the Beth Israel Deaconess Medical Center [19], where events were deaths. We extracted admissions and measurements from CareVue (ITEMID) such that admissions shared 14 measured covariates with each other, which resulted in 133 admissions. We adopted sex as a static covariate and the 14 measurements as time-varying covariates. We randomly split the individuals into 3 subgroups, assigned one to test and the others to training data, and conducted 3-fold cross evaluation of the predictive performances. The model configuration follows the experiment on PBC data set (see Appendix C).

Figure D5 displays the predictive performance on MIMIC III. It shows that SurvPP achieved better TLL performance than GBM, and SurvPP achieved comparative AUC performance with GBM. The smaller variance of SurvPP's AUC implies that SurvPP works more robustly than GBM, but further experiments are needed to investigate it.

