# OpenReview forum: "Survival Permanental Processes for Survival Analysis with Time-Varying Covariates"
_NeurIPS.cc/2023/Conference — NeurIPS 2023 poster_

### Official Review · Reviewer_feqp · 2023-06-30

**Soundness:** 4 excellent
**Presentation:** 4 excellent
**Contribution:** 4 excellent
**Rating:** 7
**Confidence:** 3

**Summary:**

The paper proposes the first coherent probabilistic formulation for survival analysis in scenarios where the covariates depend on time, present all practical details for learning MAP estimates and marginal likelihoods and predictive distributions, and provide a range of synthetic and real-data experiments that demonstrate the approach is superior to earlier heuristic solutions and standard survival models. The limitations are adequately addressed.

**Strengths:**

The paper is overall strong, providing comprehensive solution to an open problem and delivering it in a clear manner with sufficient empirical validation. I cannot immediately think of any specific way the paper could easily be improved, except perhaps by adding a clear motivational example for readers who cannot immediately understand the setting. It is by no means a perfect paper, but nevertheless represents solid scientific work in the field.

The problem is highly relevant and rarely studied. Even though some previous attempts have been made, we have been missing a proper formulation and I think that this paper provides one in a form that other can build on. The solution proposed here is definitely sound and covered well in the paper. The technical solution is not extremely novel in the sense that the hazard function is simply a transformation of a GP and standard MAP estimation is used, but this is appropriate and perhaps even ideal for a paper that proposes the formal definition of the process. It makes the paper easier to read for a broad audience, compared to one that would attempt to also use novel methodological elements within the process. The paper as written not encourages further work in improved modelling of the hazard functions and other inference techniques, and some ideas are already outlined in the conclusion.

**Weaknesses:**

The main empirical result is that we do not really observe an improvement in accuracy over the previous methods, which is naturally a bit of a disappointment. However, the proposed method is faster.

Another minor weakness is evaluation only on a fairly simple artificial data. A brief demonstration on the value of properly addressing time-varying covariates in some easy-to-understand application task would have made the paper more interesting also for audiences outside the core NeurIPS readers.

**Questions:**

1) What would be the most interesting motivational example of importance of time-varying context that you could describe in 4-5 lines, with detailed examples of what the time-varying covariates could be and what goes wrong if using standard survival models? Could you fit that in your response and then in the Introduction?

---

> ### Author Rebuttal · Authors · 2023-08-10
>
> We would like to thank the reviewer for the highly positive comments and suggestions, by which we are strongly encouraged!
>
> **What would be the most interesting motivational example of importance of time-varying context that you could describe in 4-5 lines, with detailed examples of what the time-varying covariates could be and what goes wrong if using standard survival models? Could you fit that in your response and then in the Introduction?**
>
> According to the reviewer's suggestion, we will fit the following explanation in the Introduction. We hope that our explanation is satisfactory.
>
> When exogenous and controllable covariates are considered, survival models enable us to estimate a hazard function as an explicit function of the covariates, by which we can perform *what-if analysis*, that is, simulations of the future failure events under various covariate function over time. Then we can find the optimal covariate function or policy that would yield a desirable future state, which is an important application of survival analysis with controllable covariates. Time-varying covariates are common in survival data, and if survival models for static covariates are applied to such a non-stationary data, then the static survival models fail to estimate the underlying dependence of hazard function on covariates, resulting in unreliable decision-makings. Therefore, survival models that accommodate time-varying covariates are needed. For example, in reliability engineering applications where the events are the failure times of machines, time-varying covariates could be the room temperature/humidity, which is controlled by air conditioners, and the scheduled maintenance. The maintenance manager can optimize the schedules of maintenance and air conditioning by balancing the risk of failure and the costs of maintenance and air conditioning.

---

> > ### Comment · Reviewer_feqp · 2023-08-15
> > **Reply**
> >
> > Thank you for the response. I think that example matches what I was looking for.
> >
> > More generally, I see that other reviewers pointed out related work that should be discussed in the paper, pointing out significant body of literature I was not aware of during my review. I agree with them that clarifying the exogeneous vs endogenous covariates aspect is critical. I still retain my evaluation that for a reader that is not specifically an expert on survival analysis the paper is well written and tells a convincing story and hence have no need to change my personal evaluation, but having read the other reviews I can see that there is still some effort in convincing the perhaps more important category of readers that are closely following the latest literature.

---

> > > ### Author Response · Authors · 2023-08-15
> > > **Thanks for the reply**
> > >
> > > Thank the reviewer for the kind reply to our rebuttal. As you see in other reviewers' comments, our study's position from the viewpoint of exogeneous and endogenous covariates was not clear in the original draft, which unfortunately misleads the references we should focus on or compare with our model. We suggest substantial improvements for it (our 2nd response to Reviewer 4Stf and our 4th response to Reviewer 3BPs), which we believe can address the problem. We appreciate it if you also have time to check them.

---

### Official Review · Reviewer_SVLR · 2023-07-03

**Soundness:** 4 excellent
**Presentation:** 4 excellent
**Contribution:** 3 good
**Rating:** 4
**Confidence:** 3

**Summary:**

This manuscript proposes a Gaussian process based method for survival modeling (or time-to-event modeling) assuming time-varying covariates.  Authors assume covariates are time-varying and they are treated using the counting process format which assumes covariates remain fixed until to the next measurement time point.  The hazard function (or square root of it) is assumed GP prior. Authors rely on previous work on path integral representation of GPs and derive a maximum a posterior estimator of the latent function that serves the hazard function for the likelihood/point process. The method is evaluated on a simulated data set with the log-linear as well as non-linear dependence on covariates.  On simulated data the proposed method performs on par with previous methods but has some beneficial performance in terms of computational efficiency.

**Strengths:**

Authors study the important question of time-to-event prediction in the case of time-varying covariates. Model formulation that assumes GP prior for the (square of the) hazard function and development of the methodology using path integration representation of GPs in survival modeling appear novel. Quality and clarity of the work appears very good. Problem setting is important.

**Weaknesses:**

The development of the maximum a posterior estimate of the latent function as such relies strongly on the previous work (e.g. ref. [19]) and it is unclear what part of the method development is actually novel. The proposed method is tested only on a single simulated data setting with linear and non-linear dependency on time-varying covariates.  On simulated data, performance of the model (in terms of accuracy) is only comparable relative to previous methods.  The proposed method is tested primarily on a very low-dimensional setting (2-dimensions).  Additional experiments presented in Appendix show that the performance of the method in a real data set is slightly weaker than that of the standard CoxPH model.

**Questions:**

Authors note in Conclusions section (and in Appendix) that the kernel hyper-parameters are optimised using the grid-search. Is the proposed method sensitive to kernel parameter values?  Is it due to the grid search that you need to assume the same length scales for all input features, and is that limiting the performance?

**Limitations:**

I believe authors have adequately described limitations.

---

> ### Author Rebuttal · Authors · 2023-08-09
>
> We would like to thank the reviewer for carefully reading our paper and giving valuable comments, to which we provide a detailed response below. We believe that our explanation is satisfactory.
>
> **The development of the maximum a posterior estimate of the latent function as such relies strongly on the previous work (e.g. ref. [19]) and it is unclear what part of the method development is actually novel.**
>
> Thank the reviewer for pointing out our inadequate explanation about the novelty. Although the key derivation of algorithms is based on the methodology proposed in ref. [18,19], our proposed model (SurvPP) is a non-trivial extension of the point process model proposed by ref.[19]: (i) SurvPP can accommodate multiple trials of event sequence data, while ref.[19] assumes one trial of evet sequence; (ii) in SurvPP, the end time of observation is a stochastic variable which depends on the time of event occurring, while ref.[19] assumes that the end time of observation is given. We discovered for the first time that the representer theorem, a beneficial property for functional analysis, holds for such a complicated point process, that is, survival analysis models, which is a remarkable technical contribution of our paper. We believe that the result makes a non-incremental impact on the survival analysis community.
>
> **Is the proposed method sensitive to kernel parameter values? Is it due to the grid search that you need to assume the same length scales for all input features, and is that limiting the performance?**
>
> Kernel methods/GP models are generally sensitive to kernel parameter values, and a clever way of optimizing the kernel parameter is essential. Gaussian kernels, which was used in the paper, naively require a scale parameter for each dimension of data, resulting in a high dimensional kernel parameter. The grid search becomes prohibitively costly with high dimensional parameter, but a well-known approach to the problem is to normalize (e.g., centering and scaling) each dimension of the data and put a common scale parameter across all dimensions of normalized data, which we adopted in the paper. This approach empirically works robustly, but a more sophisticated approach could improve the performance of SurvPP, which is an important next step of our study. We will add the discussion in the manuscript.
>
> **The proposed method is tested primarily on a very low-dimensional setting (2-dimensions).**
>
> In the real-world data experiment on PBC dataset, the proposed method was tested on 12-dimensional covariates, which we believe alleviates the reviewer's concern about the dimensionality of covariates. We will move it to the main paper.
>
> **Additional experiments presented in Appendix show that the performance of the method in a real data set is slightly weaker than that of the standard CoxPH model.**
>
> The result in Figure S2 that SurvPP showed slightly worse AUC than the other models could be mainly due to the inadequate optimization of kernel parameters. Figure R3 (see a pdf file submitted) plots the AUC performances on PBC dataset for SurvPPs with different search ranges of kernel parameters: the range of SurvPP_0 is $\theta \in \\{ 0.1, 0.2, 0.5, 0.7, 1.0, 2.0, 5.0, 7.0, 10.0 \\}$, which was used in the supplement; the range of SurvPP_1 is $\theta \in \\{ 0.01, 0.02, 0.03, \dots, 0.09 \\}$. Figure R3 shows that SurvPP_1 achieved substantially better AUC than SurvPP_0, and SurvPP_1's AUC is now comparable to the other models on average. More sophisticated algorithm for hyperparameter optimization could enhance the performance of SurvPP, which is the next step of our study.
>
> Also, Figure S2 shows that CoxPH achieved very high AUCs ($\gtrsim 0.9$), suggesting that the underlying generative process could be consistent with CoxPH's simple assumption of log-linearity and proportionality. Then, the semi-parametric model, CoxPH, is likely to achieve equal or better performances than the nonparametric model, SurvPP, which is consistent with the result.
>
> To test the proposed method on a more non-linear structure, we ran an additional experiment on a real-world dataset, *Standard And New Antiepileptic Drugs study data* (SANAD), provided by R package joineR (GPL-3). SANAD was an unblinded randomized trial recruiting patients with epilepsy for whom carbamazepine (CBZ) was considered to be standard treatment and they were randomized to CBZ or the newer drug lamotrigine (LTG), where 605 patients were included and event was the time to treatment failure. We adopted a time-varying covariate or calibrated dose, and three static covariates including age of patient at randomization, gender, and randomized treatment (CBZ or LTG). The way of 10-fold cross evaluation of performance follows the experiment on PBC data. Figure R2 plots the result, showing that SurvPP achieved better performance than CoxPH, and achieved comparable performance while being substantially faster than GBM.
>
> We will add the two real-world experiments in the main paper.

---

### Official Review · Reviewer_3BPs · 2023-07-06

**Soundness:** 2 fair
**Presentation:** 2 fair
**Contribution:** 1 poor
**Rating:** 5
**Confidence:** 4

**Summary:**

The paper proposes a non-parametric Bayesian survival model called Survival Permanental Process (SurvPP) for analyzing survival data with time-varying covariates. The model is based on a permanental process that defines the latent hazard function on covariate space and can handle right-censored observations in a counting process format. The paper shows that SurvPP holds a representer theorem, which offers a fast Bayesian estimation algorithm that scales linearly with the number of observed events without relying on Markov Chain Monte Carlo computation. The paper evaluates SurvPP on synthetic data and shows that it achieves comparable predictive accuracy while being hundreds of times faster than state-of-the-art methods (specifically, CoxPH and GBMs).

**Strengths:**

The main strengths of this paper are listed below:

1. The paper introduces a novel non-parametric model, called SurvPP, for estimating hazard functions based on survival data with time-varying covariates. This model addresses the limitation of the Cox proportional hazards model (CoxPH) by allowing for non-linear dependence between survival times and covariates, and also where these covariates can change, after the time when the predictions are made.

2. The paper shows that SurvPP holds a representer theorem, which means it can be trained efficiently using a fast Bayesian estimation algorithm that scales linearly with the number of observed events without relying on Markov Chain Monte Carlo computation.

3. The authors have conducted experiments on two synthetic datasets to demonstrate that their proposed SurvPP outperforms CoxPH and another non-parametric model called Generalized Boosted Regression Models (GBM) in terms of predictive performance. SurvPP achieves significantly better performance than CoxPH, but the performance gaps between SurvPP and GBM are marginal and not significant.

4. The paper shows that SurvPP is computationally efficient, particularly when covariates are measured frequently. SurvPP can be trained faster than GBM and CoxPH in estimating hazard functions when covariates are measured frequently.

**Weaknesses:**

(See next entry: Questions )

**Questions:**

Although this paper is well written and technically sound, the reviewer believes that the paper has very limited practical utility. Below are the points that seem serious limitations, which the authors need to address:

**1:** While the paper presents a novel approach to survival analysis using permanental processes and provides a comparison with conventional models (Cox-PH and GBM), the experimental validation appears to be limited to synthetic datasets with only two-dimensional covariates. The reviewer would like to see how the proposed model performs on real-world datasets, which often have higher dimensionality and more complex structures. Specifically, it would be beneficial to test the model on widely recognized and publicly available datasets such as the MIMIC III (Medical Information Mart for Intensive Care III) or the U.K. Cystic Fibrosis Registry. These datasets contain a rich variety of covariates (that do vary over time) and present real-world challenges that synthetic datasets may not fully capture. Using such datasets would not only provide a more rigorous test of the model's performance but also make the results more relatable to practitioners in the field. It would also allow for a more direct comparison with other models in the literature that have been tested on these datasets.

The authors should explain why they chose to use these synthetic datasets for their experiments, by motivating the particular covariate functions they used–Equations 26, 27, etc.--explaining why they match a realistic situation. Additionally, they should extend the experimental validation to include real-world datasets, to further demonstrate the applicability and robustness of the proposed model.  This would address some other concerns–eg, how to determine the appropriate kernel to use, etc.

**2:** Additionally, the scalability of the proposed model in the context of high-dimensional, low-sample size problems remains an area that warrants further exploration. While we see the authors have acknowledged this limitation (Section 5, lines 308 onwards), the current treatment of this issue does not fully address the potential implications for the practical application of the proposed model. In real-world scenarios, datasets often exhibit high dimensionality, and it is common to encounter situations where the sample size is relatively small. These conditions pose significant challenges for many machine learning models, and it is crucial to understand how the proposed model would perform under such circumstances.The Bayesian estimation framework proposed in this paper adds another layer of complexity to this issue.. Thus, a more detailed discussion of how the proposed Bayesian estimation framework would scale with data dimensionality would be highly beneficial. Therefore, the authors are advised to expand their discussion on this topic. Specifically, it would be valuable to include theoretical considerations, potential strategies for handling high-dimensional data, and, if possible, empirical results from preliminary experiments with (maybe synthetic) high-dimensional, low-sample size datasets.

In addition to a discussion about the sample efficiency, the paper should also discuss the learner’s computational efficiency.

**3:** The performance of the proposed model is compared with two benchmark methods, namely the Cox proportional hazards model (CoxPH) and the generalized boosted regression model (GBM). While these comparisons provide valuable insights into the model's performance, the choice of benchmark methods raises some questions. Specifically, the CoxPH and GBM models, although well-established in the field, do not represent the most recent advancements in survival analysis. It would be interesting to see how the proposed model compares with more recent methods that leverage advanced machine learning techniques. Specifically, the authors should compare their approach with the following methods:

*Nagpal, Chirag, Vincent Jeanselme, and Artur Dubrawski. "Deep parametric time-to-event regression with time-varying covariates." Survival Prediction-Algorithms, Challenges and Applications. PMLR, 2021.*

*C. Lee, J. Yoon and M. v. d. Schaar, "Dynamic-DeepHit: A Deep Learning Approach for Dynamic Survival Analysis With Competing Risks Based on Longitudinal Data," in IEEE Transactions on Biomedical Engineering, vol. 67, no. 1, pp. 122-133, Jan. 2020, doi: 10.1109/TBME.2019.2909027.*

*Tomašev, Nenad, et al. "Use of deep learning to develop continuous-risk models for adverse event prediction from electronic health records." Nature Protocols 16.6 (2021): 2765-2787.*

Comparing the proposed model with these more recent methods would provide a more comprehensive evaluation of its performance and relevance in the current research landscape. It would also allow for a more direct comparison with state-of-the-art models, enhancing the paper's contribution to the field.

**4:** The authors have presented their experimental results in a cross-validation setting for synthetic datasets. While cross-validation is a widely accepted method for model evaluation, particularly when dealing with limited data, it is not immediately clear why this approach was chosen for synthetic data, where the volume of data can be controlled. Synthetic data offers the unique advantage of being able to generate as much data as needed, which could potentially allow for the creation of independent validation sets. This approach would provide a more stringent test of the model's generalization ability, as it would involve testing on data that the model has not seen during training, even indirectly through cross-validation folds.

Could the authors provide some insights into their decision to use cross-validation for synthetic data? Furthermore, would it be possible to conduct additional experiments using independent validation sets with synthetic data to further assess the model performance?

---

> ### Author Rebuttal · Authors · 2023-08-09
>
> We would like to thank the reviewer for carefully reading our paper and giving valuable comments, to which we provide a detailed response below. We believe that we can fully address all the concerns.
>
> **1: the experimental validation appears to be limited to synthetic datasets with only two-dimensional covariates. The reviewer would like to see how the proposed model performs on real-world datasets, which often have higher dimensionality and more complex structures....**
>
> We totally agree with the reviewer about the importance of validation on real-world datasets. Actually, we performed an experiment on a real-world dataset (PBC dataset), but put the result in the supplement (Section S4) due to space limitation. We apologize for the inconvenience. According to the reviewer's suggestion, we added one more experiment on another real-world data, and will put in the main paper the following two real data experiments.
>
> PBC dataset has twelve covariates, which we believe alleviates the reviewer's concern about the dimensionality of covariates. But Figure S2 in the supplement shows that CoxPH achieved very high AUCs ($\gtrsim 0.9$), suggesting that the underlying generative process could be consistent with CoxPH's simple assumption of log-linearity and proportionality. Then, the semi-parametric model, CoxPH, is likely to achieve equal or better performances than the nonparametric model, SurvPP, being consistent with the result.
>
> This time, we ran an additional experiment on a real-world dataset, *Standard And New Antiepileptic Drugs study data* (SANAD), provided by R package joineR (GPL-3). SANAD was an unblinded randomized trial recruiting patients with epilepsy for whom carbamazepine (CBZ) was considered to be standard treatment and they were randomized to CBZ or the newer drug lamotrigine (LTG), where 605 patients were included and event was the time to treatment failure. We adopted a time-varying covariate (calibrated dose), and three static covariates (age of patient, gender, and randomized treatment (CBZ or LTG)). The procedure of 10-fold cross evaluation of performance follows that of the experiment on PBC data. Figure R2 plots the result, showing that SurvPP achieved better performance than CoxPH, and achieved comparable performance while being substantially faster than GBM.
>
> MIMIC III and U.K. Cystic Fibrosis Registry, kindly suggested by the reviewer, seem to require more than a week for access approval. Thus, we gave up testing them in the rebuttal period. We hope that it is acceptable to the reviewer.
>
> **The authors should explain why they chose to use these synthetic datasets for their experiments, by motivating the particular covariate functions they used–Equations 26, 27, etc...**
>
> We prepared the synthetic datasets (Eq.(26-27)) for evaluating the methods' predictive and time performances in the condition that time-varying covariates have linear and nonlinear effects on the hazard function. Synthetic data with similar nonlinearity has been adopted in the literature [g], and we believe that the synthetic datasets are in line with the purpose of this paper.
>
> [g] Katzman et al. DeepSurv: personalized treatment recommender system using a Cox proportional hazards deep neural network. BMC medical research methodology, 18(1):1-12, 2018.
>
> **2: the scalability of the proposed model in the context of high-dimensional, low-sample size problems remains an area that warrants further exploration...Thus, a more detailed discussion of how the proposed Bayesian estimation framework would scale with data dimensionality would be highly beneficial.**
>
> Thank the reviewer for an important point of view! One of the biggest advantages of kernel method/GP is the good scalability with data dimensionality, $d$, from which SurvPP have benefitted fully: the complexity of SurvPP regarding $d$ is $\mathcal{O}(NMQ+(N+J)M^2+M^3+dJM)$, where $\mathcal{O}(dJM)$ stems from the computation of feature maps, $\phi_m(\mathbf{y})$. As is $J$, $d$ is not part of the iterative optimization ($Q$), resulting in a very feasible computation regarding the data dimensionality. We will add the discussion in the main paper, which highlights the merit of our model.
>
> **3: the choice of benchmark methods raises some questions. Specifically, the CoxPH and GBM models, although well-established in the field, do not represent the most recent advancements in survival analysis...**
>
> All the references raised by the reviewer are joint modeling approaches for endogenous covariates, while our SurvPP is for exogeneous covariates. Please see our 2nd response to Reviewer 4Stf about the difference between endogenous and exogenous covariates: SurvPP cannot be compared directly with the approaches for endogenous covariates because the tasks considered are different with each other. Also, it is notable that as well as CoxPH and GBM, we adopted RFM as a benchmark, which is one of the most recent advancements in survival analysis for exogenous covariates published in 2020 (arXiv) and in 2021 (journal).
>
> **4: it is not immediately clear why this approach (cross-validation) was chosen for synthetic data...Could the authors provide some insights into their decision to use cross-validation for synthetic data? Furthermore, would it be possible to conduct additional experiments using independent validation sets with synthetic data to further assess the model performance?**
>
> We chose the cross-validation approach just because it is common in the machine learning literature. We are pleased to follow the reviewer's suggestion. Figure R4 displays the result on independent validation sets of synthetic data ($U$ = 900 for each of 10 training data, and $U$ = 100 for each of 10 test data), showing the result similar to the cross-validation approach (Figure 1-2). We will add the result in the supplement. If the reviewer is interested in more large samples or trials, we promise to add experiments in the final manuscript.

---

> > ### Comment · Reviewer_3BPs · 2023-08-21
> > **Many issues resolved ... but two more issue,**
> >
> > Although the authors have suggested significant modifications to their paper in the rebuttal, there are still two issues that need to be addressed:
> >
> > 1. The revised experiment plan with real-world datasets seems great, but it still includes datasets with a limited number of input features. It would be great if authors could include some experiments with suggested datasets (MIMIC III, etc.) to demonstrate the effectiveness of the proposed approach on high-dimensional low-sample-size datasets. If this is not possible, then please mention this in the limitations or future work section of the paper.
> >
> > 2. Computational complexity analysis needs to be more detailed to specifically explain the scalability of the proposed approach with respect to the number of input features (with respect to modern-day massive datasets generated through 'omics technologies, etc.)
> >
> > Another comparative method for exogenous variables could be the following one.
> >    https://www.sciencedirect.com/science/article/pii/S0377221722008104
> >
> > Given the improvements, and the assurances of future work, I am increasing my score.

---

> > > ### Author Response · Authors · 2023-08-21
> > > **Thanks for the reply**
> > >
> > > We thank the reviewer a lot for reading our rebuttal and giving valuable comments! Also, we appreciate the information about the most recent reference (published just a month after we submitted this NeurIPS paper) for exogenous time-varying covariates, which we will cite in Related Work with respect.
> > >
> > > **1st issue raised by reviewer**
> > >
> > > We promise to add in the final draft an experiment with MIMIC III or U.K. Cystic Fibrosis Registry (maybe, in the supplement), while we will mention in the limitations the need for model validations with more real-world high-dimensional low-sample-size datasets.
> > >
> > > **2nd issue raised by reviewer**
> > >
> > > We thought that our response to your comment ("2: the scalability of the proposed model in...") could address the question about the scalability of the proposed approach with respect to the number of input features (denoted by $d$), but we might misunderstand your question. Sorry for the inconvenience, but could you please elaborate on what kind of complexity analysis you want?

---

### Official Review · Reviewer_LBpm · 2023-07-07

**Soundness:** 3 good
**Presentation:** 3 good
**Contribution:** 3 good
**Rating:** 7
**Confidence:** 3

**Summary:**

This paper addresses a survival analysis problem with time-varying covariates by proposing a novel Bayesian approach to estimate a hazard function as a nonlinear function of covariates using the kernel trick. MAP, predictive distribution, and computational complexity of the proposed approach were presented. Experiments were done on both synthetic and real-world benchmark datasets.

**Strengths:**

- Well-written and easy to follow. The relationship with the related work is clearly presented.

- Proposed a novel Bayesian approach to tackle the survival analysis with time-varying covariates that allows modeling of a nonlinear relationship between hazard and covariates. The authors presented MAP, predictive distribution, and computational complexity of the proposed approach.

- Experimental results clearly support that SurvPP provides good predictive performance as well as is computationally efficient.

**Weaknesses:**

- Just as other GP or kernel-based methods, SurvPP might not be ideal for complex high-dimensional data. But the authors have pointed this out, and a potential way to handle this has been clearly discussed.

- In the experimental results on the real-world benchmark dataset, SurvPP showed slightly worse AUC than the other models, unlike in synthetic data. Could the authors kindly give some explanation why? Interestingly, in terms of TLL, SurvPP is doing quite well.


**Questions:**

- The authors might not have enough space to put the experimental results on the real-world benchmark dataset. Would the authors plan to move it to the manuscript?

- Is there any specific reason that the authors used AUC rather than C-index, which is more commonly used in the survival analysis?

- It is interesting that SurvPP, which is a non-parametric model, showed as good TLL as CoxPH, which is a (semi) parametric model, unlike the other nonparametric tree-based approaches. Could authors give some explanation why this happened? Is this something expected?

- I am also curious about what each method's survival curve looks like. Might be interesting if the authors could compare the average prediction curve against the KM survival curve.

- As the proposed method is a Bayesian approach, it can provide more than just better prediction performance. Maybe the authors could add some discussion on that.

**Limitations:**

Did not find any other limitations than the authors discussed in the last section.

---

> ### Author Rebuttal · Authors · 2023-08-09
>
> We would like to thank the reviewer for the highly positive comments and suggestions, by which we are strongly encouraged. Below we provide a detailed response to each of the comments.
>
> **The authors might not have enough space to put the experimental results on the real-world benchmark dataset. Would the authors plan to move it to the manuscript?**
>
> Yes, we will put in the main paper the real data experiment. Furthermore, we added one more experiment on real-world data (for details, see Figure R2 and our response to Reviewer LBpm).  We will use an extra page given for accepted papers. Sorry for the inconvenience.
>
> **Is there any specific reason that the authors used AUC rather than C-index, which is more commonly used in the survival analysis?**
>
> C-index considers the concordance between the ranking of the test results and that of the observed times-to-events, where the ranking of the test results is assumed to be static. It is not appropriate as an evaluation criterion when covariates are time-varying because the ranking of the test results can also be time-varying. As is clear from the definition in the paper (Eq.(28)), the dynamic AUC essentially estimates the C-index at each time, and is commonly used in the literature of survival analysis with time-varying covariates. Note that it is known that a weighted average of the dynamic AUC equals C-index [e]. We found that our introduction was inadequate from the reviewer comment, and will add the above explanation in the main paper.
>
> [e] Pantoja-Galicia et al. Concordance measures and time-dependent ROC methods. Biostatistics & Epidemiology 5(2): 232-249, 2021.
>
> **In the experimental results on the real-world benchmark dataset, SurvPP showed slightly worse AUC than the other models, unlike in synthetic data. Could the authors kindly give some explanation why?**
>
> The result that SurvPP showed slightly worse AUC than the other models could be mainly due to the inadequate optimization of kernel parameters. Figure R3 (see a pdf file submitted) plots the AUC performances on PBC dataset for SurvPPs with different search ranges of kernel parameters: the range of SurvPP_0 is $\theta \in \\{ 0.1, 0.2, 0.5, 0.7, 1.0, 2.0, 5.0, 7.0, 10.0 \\}$, which was used in the supplement; the range of SurvPP_1 is $\theta \in \\{ 0.01, 0.02, 0.03, \dots, 0.09 \\}$. Figure R3 shows that SurvPP_1 achieved substantially better AUC than SurvPP_0, and SurvPP_1's AUC is now comparable to the other models on average. More sophisticated algorithm for hyperparameter optimization could enhance the performance of SurvPP, which is the next step of our study. We will add the above discussion in the supplement.
>
> **It is interesting that SurvPP, which is a non-parametric model, showed as good TLL as CoxPH, which is a (semi) parametric model, unlike the other nonparametric tree-based approaches. Could authors give some explanation why this happened? Is this something expected?**
>
> Thank the reviewer for the good point of view. SurvPP treats the duration from the entry, $t$, as a time-varying covariate, and estimates the hazard function of $t$ with the smoothness specified by the kernel function. But the tree-based approaches, GBM and RFM, estimate the base hazard function $h(t)$ (Eq.(25)) by the Breslow/Fleming-Harrington estimator, where $h(t)$ is treated as piecewise constant between uncensored failure times [f]. Therefore, if the underlying base hazard function is smooth and the number of observed failure times is small, then GBM and RFM could estimate the hazard function regarding $t$ poorly compared to SurvPP. TLL evaluates the base hazard function directly, while AUC disregards the contribution of the base hazard function common to all individuals, which is a possible reason of the obtained results.
>
> Also, CoxPH's very high AUC ($\gtrsim 0.9$) in PBC dataset suggests that the underlying generative process could be consistent with the log-linear and proportional assumptions of CoxPH, in which the semi-parametric model, CoxPH, is likely to achieve equal or better performances than the nonparametric model, SurvPP. As a response to another reviewer, we added one more experiment on real-world data, showing that SurvPP performed better than CoxPH when the log-linear assumption is not valid (for details, see Figure R2 and our response to Reviewer 3BPs).
>
> We will add the detailed discussions about the experimental results in the main paper.
>
> [f] Lin. On the Breslow estimator. Lifetime data analysis 13:471-480, 2007.
>
> **As the proposed method is a Bayesian approach, it can provide more than just better prediction performance. Maybe the authors could add some discussion on that.**
>
> Thank the reviewer for a good suggestion. We will add the following discussion.
>
> One of the advantages of Bayesian approaches over non-Bayesian ones is that they enable us to perform uncertainty evaluations. More concretely, the posterior process of SurvPP is also given by a permanental process, and thus we can generate sample paths of the estimated hazard function of covariates, $\lambda (\mathbf{y})$. Using the samples, we can evaluate the distribution of survival function and perform risk-aware survival analyses.

---

> > ### Comment · Reviewer_LBpm · 2023-08-20
> >
> > I want to thank the authors for their clarification and feedback. I have carefully gone through the feedback and discussions with other reviewers. I agree that the authors need to clarify the exogenous and endogenous aspects, which they agreed to discuss in the revised manuscript. But other than that, I have not found a critical reason to change my score.

---

> > > ### Author Response · Authors · 2023-08-21
> > > **Thanks for the reply**
> > >
> > > We thank the reviewer a lot for having time to read not only our rebuttal for you but also those for other reviewers, while for acknowledging it! We really appreciate replies to our rebuttals, regardless of whether they agree or disagree with our rebuttals.

---

### Official Review · Reviewer_4Stf · 2023-07-08

**Soundness:** 3 good
**Presentation:** 2 fair
**Contribution:** 2 fair
**Rating:** 3
**Confidence:** 4

**Summary:**

This paper introduces a new hazard function in survival analysis. The hazard is a squared Gaussian process, where the index set is the covariates at a given time. They assume that the covariates are piecewise constant, propose a MAP estimator and derive the predictive distribution and marginal likelihood. They use ideas from kernel methods and achieve computational complexities that are lower than alternate approaches, scaling linearly with the number of observed events.

**Strengths:**

The idea of connecting this to kernel methods to reduce computational complexity is quite clever, and the speedups over GBM and RFM are impressive.

**Weaknesses:**

The piecewise constant time-varying covariate assumption is very strong. You should mention this.

There also needs to be some discussion of exogeneous vs endogeneous time-varying covariates. It looks like you're implicitly assuming that your covariates are exogeneous, but you should make that clear.

You need a real data experiment in the main paper. In the introduction, you mention 'confirm that our algorithm achieves comparable pre67 dictive accuracy while being hundreds of times faster than state-of-the-art survival models' but saying nothing about real data. You have something in the supplement, but again, there's no reference to how good the results are in the main paper. Given that, one has to give less weight to the supplement.

In your benchmarks, I assume that all of them are assuming piecewise constant time-varying covariates. You should include something using interpolation: e.g. linear mixed models with splines. They do this a lot in the joint modeling literature.

**Questions:**

Are your benchmarks also assuming piecewise constant time-varying covariates?

**Limitations:**

While the limitation they mention are important, some of the limitations that I mentioned in the weaknesses are not addressed.

---

> ### Author Rebuttal · Authors · 2023-08-08
>
> We would like to thank the reviewer for the comment, to which we provide a detailed response below. Especially, the critical concern about the piecewise constant assumption of covariates arises from our confusing explanations or introductions, and we believe that we can fully dispel the concern. We hope that our explanation is satisfactory.
>
> **The piecewise constant time-varying covariate assumption is very strong. You should mention this.**
> **You should include something using interpolation: e.g. linear mixed models with splines. They do this a lot in the joint modeling literature.**
> **Are your benchmarks also assuming piecewise constant time-varying covariates?**
>
> The piecewise constant time-varying covariate assumption, which is referred as the *counting process format of input* in the paper, approximates the continuous covariate function over time by a set of representative points (RPs) at finite time points. There are two ways of defining the RPs: one is as the points that were observed, which might be sparse over time due to measurement constraints ; the other is as the denser points obtained by using interpolation methods (e.g., splines). We think that the reviewer assumes the former definition and concludes that the covariate function should be approximated by denser points. But our model (SurvPP) and the benchmarks (CoxPH, GBM, RFM) assume the latter definition in the paper, and thus various numbers of splitting ($J_u$) were considered in the synthetic data experiments. Therefore, the counting process format of input is not so strong because we can adopt RPs in any density if necessary. Note that our model achieves an efficient computation regarding $J_u$ (see Figure 1-2).
>
> Also, we consider the possibility that the reviewer suggests us to utilize the functional form of interpolated covariate function, without discretizing the interpolated covariate function. The suggestion makes sense, but unfortunately, we cannot find such approaches in the joint modeling literature: covariate function needs to be discretized over time in RNN-based approaches [a,b]; in Cox proportional hazards-based models, the non-parametric estimator of baseline intensity is zero except at observed event times, which results in the discretization of covariate function for likelihood function evaluation [c]. However, thanks to the reviewer's suggestion, we found that our proposed algorithm could be improved as follows. The functional form of (interpolated) covariate function involves the sum of outer products of feature maps, denoted by $\mathbf{A}$ in Eq. (16), and it can be considered as a Riemann sum approximation of the time integral,
>
> $$
> \mathbf{A} = \int_0^T \phi(\mathbf{y}(t)) \phi(\mathbf{y}(t))^{\top} dt \sim \sum_{j=1}^J \Delta_j \mathbf{\phi}(\bf{y}_j)\mathbf{\phi}(\mathbf{y}_j)^{\top}.
> $$
>
> Given a smooth covariate function, the Riemann sum approximation can be replaced by a more accurate approximation,
>
> $$
> \mathbf{A} = \int_0^T \phi(\mathbf{y}(t)) \phi(\mathbf{y}(t))^{\top} dt \sim \sum_{j=1}^J w_j \mathbf{\phi}(\mathbf{y}_j)\mathbf{\phi}(\mathbf{y}_j)^{\top},
> $$
>
> where the weight, $w_j$, depends on the approximation rule. This time, we adopt the Simpson's 1/3 rule, and add an experiment with the synthetic nonlinear data to examine the improvement in accuracy. Note that the Simpson's 1/3 rule corresponds to a quadratic interpolation. Figure R1 (see a pdf file submitted) shows that adopting the Simpson's rule achieved a better predictive performance in TLL with a smaller discretization number ($J_u$). We will add the above discussions and results in the manuscript. We appreciate the reviewer's valuable comment.
>
> **There also needs to be some discussion of exogeneous vs endogenous time-varying covariates. It looks like you're implicitly assuming that your covariates are exogeneous, but you should make that clear.**
>
> As the reviewer pointed out, we consider exogenous covariates in the paper. Although we discussed about the topic in Related Work (Two Scenarios in ...), we found that the discussion was inadequate. According to the reviewer's suggestion, we will revise the paragraph to explain the following points: the definitions of exogenous and endogenous covariates [d]; the primary purpose of survival analysis with exogenous covariates is to estimate a hazard function as an explicit function of covariates, and to make predictions of failure times under various possible *future* covariate functions (what-if analysis); the primary purpose of survival analysis with endogenous covariates is to make predictions of failure times by using the *past* observations of covariates, where joint modeling approaches, which jointly model the stochastic process of covariates and failure times, have been developed intensively; we consider exogenous covariates, and survival models for exogenous covariates cannot be compared directly with those for endogenous covariates because the tasks are different with each other.
>
> **You need a real data experiment in the main paper.**
>
> According to the reviewer's suggestion, we will put in the main paper the real data experiment (now in the supplement). Furthermore, we added one more experiment on real-world data (for details, see Figure R2 and our response to Reviewer LBpm).  We will use an extra page given for accepted papers.
>
> [a] Lee et al. Dynamic-DeepHit: A deep learning approach for dynamic survival analysis with competing risks based on longitudinal data. IEEE Transactions on Biomedical Engineering, 67(1):122–133, 2019.
> [b] Nagpal et al. Deep parametric time-to-event regression with time-varying covariates. In Survival Prediction-Algorithms, Challenges and Applications, 184–193, 2021.
> [c] Jarrett et al. Dynamic prediction in clinical survival analysis using temporal convolutional networks. IEEE Journal of Biomedical and Health Informatics, 24(2):424–436, 2019.
> [d] Kalbfleisch and Prentice. The Statistical Analysis of Failure Time Data. John Wiley & Sons, 2011.

---

### Author Rebuttal · Authors · 2023-08-10

We would like to thank all reviewers for valuable comments. We provided responses in as much detail as possible. We believe that we can fully dispel the concerns raised by the reviewers. Please see the submitted pdf file which includes newly added figures (Figure R1, R2, R3, R4) as responses to the reviewers' suggestions.

---

### Decision · Program_Chairs · 2023-09-21

**Decision:**

Accept (poster)

**Comment:**

This well-written paper has been assessed by five knowledgeable reviewers who varied in their opinions: two recommended straight acceptance, one borderline acceptance, one borderline rejection, and one straight rejection. The work proposes new nonparametric hazard estimator for survival analysis, supported by empirical studies and theoretical discussion. Most of the concerns raised by the reviewers have been addressed by the authors' rebuttals, however the two reviewers who issued the lowest scores did not participate in the discussion. Overall assessment is that this work is sufficiently novel to attract attention of the NeurIPS audience and should be accepted. The authors are urged to reflect the key points raised in the discussion with the reviewers in the final version of their manuscript.